

# Evaluation of the reflectivity calibration of W-band radars based on observations in rain

Alexander Myagkov[1], Stefan Kneifel[2], and Thomas Rose[1]

[1]Radiometer Physics GmbH, Meckenheim, Germany
[2]Institute for Geophysics and Meteorology, University of Cologne, Cologne, Germany

**Correspondence:** Alexander Myagkov (alexander.myagkov@radiometer-physics.de)

**Abstract.** This study presents two methods to evaluate the reflectivity calibration of W-band cloud radars. Both methods use natural rain as a reference target. The first approach is based on a self-consistency method of polarimetric radar variables, which is widely used in the precipitation radar community. As previous studies pointed out, the method cannot be directly applied to higher frequencies, where non-Rayleigh scattering effects and attenuation have a non-negligible influence on radar variables. The method presented here solves this problem by using polarimetric Doppler spectra to separate backscattering and propagational effects. New fits between the separated radar variables allow to estimate the absolute radar calibration using a minimization technique. The main advantage of the self-consistency method is its less dependence on the spatial variability in radar drop-size-distribution (DSD). The estimated uncertainty of the method is $\pm 0.7$ dB. The method was applied to three intense precipitation events and the retrieved reflectivity offsets were within the estimated uncertainty range. The second method is an improvement of the conventional disdrometer-based approach, where reflectivity from the lowest range gate is compared to simulated reflectivity using surface disdrometer observations. The improved method corrects first for the time-lag between surface DSD observations and the radar measurements at a certain range. In addition, the effect of evaporation of raindrops on their way towards the surface is mitigated. The disdrometer-based method was applied to 12 rain events observed by vertically-pointed W-band radar and showed repeatable estimates of the reflectivity offsets at rain rates below 4 mm h$^{-1}$ within $\pm 0.9$ dB. The proposed approaches can analogously be extended to Ka-band radars. Although very different in terms of complexity, both methods extend existing radar calibration evaluation approaches, which are inevitably needed for the growing cloud radar networks in order to provide high-quality radar observation to the atmospheric community.

## 1 Introduction

During the last decades, millimeter-wavelength radars (also known as cloud radars) became an invaluable source of information for cloud and precipitation research (Kollias et al., 2007). Due to the shorter wavelengths, cloud radars are not only sensitive to precipitating particles but also to cloud droplets, small ice particles, and fog. This makes these instruments extremely valuable tools to study, for example, cloud formation, cloud microphysical processes, or the associated radiative effect of clouds. Consequently, cloud radars have been set up around the world. The U.S. Department of Energy (DOE) Atmospheric Radiation Measurement (ARM) program maintains a number of fixed stations and mobile platforms equipped with 35 and 94





GHz radars (Mather and Voyles, 2013). In Europe, many universities and atmospheric research centers have deployed cloud radars (Haeffelin et al., 2005; Illingworth et al., 2007; Bouniol et al., 2010; Löhnert et al., 2015; Hirsikko et al., 2014). The majority of cloud radars sites provide their data to the CloudNet project (Illingworth et al., 2007), which is part of the European research infrastructure ACTRIS (Aerosols, Clouds, and Trace gases Research InfraStructure; http://www.actris.eu, last access: January 2019). Cloudnet provides algorithms to produce cloud and precipitation classification and the datasets are converted
in a unified format. This allows to derive long-term cloud statistics, which of course rely strongly on proper radar calibration.

At CloudNet and ARM sites, cloud radars are often operated in collocation with a microwave radiometer and a lidar in order to derive macrophysical (Wang and Sassen, 2001; Shupe et al., 2011), microphysical (Matrosov et al., 1998; Shupe et al., 2006; Shupe, 2011; Bühl et al., 2016; Kalesse et al., 2016; Acquistapace et al., 2017) and dynamical properties (Shupe et al., 2008; Bühl et al., 2015; Borque et al., 2016; Radenz et al., 2018) of clouds.

One of the most widely used radar observables is the equivalent radar reflectivity factor (henceforth called reflectivity). This parameter depends on size, concentration, phase, shape, density, and orientation of particles. Many operational cloud property retrievals (Matrosov, 1997, 1999; Frisch et al., 2002; Hogan et al., 2006; Heymsfield et al., 2008) rely on accurate measurements of reflectivity. Some studies combine reflectivities at different frequencies in order to derive detailed microphysics of cloud particles (Matrosov, 2011; Leinonen et al., 2013; Kneifel et al., 2015) and, therefore, require precise calibration of all radar
systems involved. For networks of cloud radars, which are supposed to provide long-term observations of cloud properties, methods to validate the quality of the reflectivity calibration are of key importance. This study focuses on the reflectivity calibration, because it is one of the most commonly used parameters for retrievals and for model evaluation. This, nevertheless, does not imply that the calibration of Doppler and polarimetric observations is of less importance. Aspects of antenna pointing calibration, which is essential for accurate Doppler measurements, can be found in Huuskonen and Holleman (2007) and Muth
et al. (2012). Moisseev et al. (2002) and Myagkov et al. (2015, 2016a) showed the calibration of polarimetric variables for cloud radars operating in different configurations.

Proper calibration and monitoring of reflectivity calibration are key considering the growing number of meteorological radars worldwide. However, even radars operated within large observational networks have been shown to be sometimes prone to calibration errors (Protat et al., 2011; Ewald et al., 2019; Maahn et al., 2019; Kollias et al., 2019).

Chandrasekar et al. (2015) compiled a detailed review of the centimeter-wavelength radar calibration techniques for an operational use. Many of the described aspects are also relevant for cloud radars. Maintaining the accurate reflectivity measurements requires temperature stabilization of the radar housing, protecting antennas and radomes from water (Hogan et al., 2003; Delanoë et al., 2016), frequent automatic internal calibrations, and regular maintenance (Chandrasekar et al., 2015). Cloud radar manufacturers typically apply a budget calibration, i.e. characterize individual radar components separately during
manufacturing and take the results into account in the reflectivity calculations (Görsdorf et al., 2015; Küchler et al., 2017; Ewald et al., 2019). The budget calibration has several shortcomings. First, it requires calibrated measurement equipment and experienced technical staff. Second, during the calibration, the analyzed radar is out of operation and has to be partly disassembled. Third, the calibration accuracy still depends on the component stability during operation. Finally, the calibration procedure may significantly differ for radars of different types, which is problematic for operational cloud radar networks.





A calibration using an external target with known properties (known as end-to-end calibration) allows for mitigating of the above-mentioned problems. One of the conventional external calibration methods of meteorological radars is based on point target observations (Chandrasekar et al., 2015). Unfortunately, its applicability to cloud radars is often limited. The target has to be mounted in the far field on a tower that is often not available. For precision pointing of the radar to the target, this method requires a scanning unit, which many of the currently deployed cloud radars are not equipped with. In principle, the target
can be lifted up by a balloon or a drone. But it is challenging to achieve perfect pointing and spatial stability of the target. Both aspects are very critical due to the narrow antenna beams. In addition, as pointed out by Gorgucci et al. (1992) and Chandrasekar et al. (2015), the calibration with a point target does not directly take into account the volumetric scattering by clouds and precipitation.

      Another approach is a comparison of observations of an inspected radar against a calibrated reference system. For instance,
Protat et al. (2011) proposed to compare observations in ice clouds by ground-based cloud radars and the space-borne W-band radar CloudSat (Stephens et al., 2008). Based on the scattering from the sea surface, CloudSat reflectivity calibration is performed on a monthly basis and is accurate within ±0.5 dB (Tanelli et al., 2008). Due to the high velocity and 1.5 km footprint, direct comparison of the reflectivity value from CloudSat and a static ground-based leads to large uncertainties (standard deviation of 2-3 dB). In order to reduce the uncertainties, Protat et al. (2009) used time periods in the order of
several months for the statistical comparisons. With the CloudSat flight cycle of 16 days and the requirement in pure ice non-precipitating clouds during an overpass, this method is mainly applicable for long-term calibration monitoring (Kollias et al., 2019).

      Using natural volume-distributed targets for the calibration verification is a well established approach. The use of rain drops as reference targets allows to directly account for the antenna properties in the calibration procedure. First successful attempts
to evaluate meteorological radars with rain date back to 1968 (Atlas, 2002). Since then, several different approaches have been developed. Among the most widely used methods is the one based on disdrometer observations. Drop-size-distributions (DSD) observed in-situ are converted to the radar reflectivity. Time series (Gage et al., 2004; Frech et al., 2017) or distributions (Kollias et al., 2019; Dias Neto et al., 2019) of calculated and observed reflectivities are then compared. Hogan et al. (2003) showed a calibration verification method suitable for W-band cloud radars only. They found that for a range of DSD, the reflectivity is
about 19 dBZe for rain intensities from 5 to 20 mm/h at the range of 250 m from the radar. Clearly, one of the main sources of uncertainties for the methods using in-situ rain observations is the vertical variability of rain properties. This variability might originate from a number of effects among which are turbulence, wind shear, evaporation, drop break up, and coalescence.

      Goddard et al. (1994) proposed a self-consistency calibration method based on polarimetric radar observations at low elevation angles. Within this method radar range bins are analyzed independently and, therefore, the methods is less sensitive to
spatial variability of DSD. The method has been operationally used for the 3-GHz Chilbolton radar Hogan et al. (2003) and is well established in the weather radar community. Hogan et al. (2003) claim the accuracy of this method to be better than ±0.5 dB. Nevertheless, the authors pointed out that the method cannot be directly used for cloud radar calibration because of strong attenuation at millimeter wavelengths by liquid water and non-Rayleigh scattering effects.



This study presents two methods to evaluate the reflectivity calibration of W-Band radars. The first approach is a new attempt
to extend the polarimetric consistency method of Goddard et al. (1994) for cloud radars. Due to lower costs, polarimetric
cloud radars become increasingly available and therefore, it is highly desirable to utilize their polarization capabilities for
calibration monitoring. The second approach is an improvement of the conventional disdrometer-based method using additional
corrections for wind shear and evaporation. This method, which does not require scanning or polarimetric capabilities, is
applicable to a large number of radar sites, which are already equipped with disdrometers.

The paper is organized as follows. In Sec. 2 the used instrumentation is described. The calibration methods and their com-
parison are shown in Sec. 3. In Section 5 we summarize the estimated accuracy and discuss the applicability of the calibration
evaluation methods.

## 2 Data and instrumentation

For the comparison of different calibration methods, we combine observations from two sites. In this way, we are able to
collect a dataset with a wide range of rainfall rates observed with various radar and in-situ instrumentation. During summer
2018, a number of convective rainfall events were recorded at the Radiometer Physics GmbH (RPG) site in Meckenheim,
Germany (henceforth RPG site). The site is equipped with a demonstration W-Band cloud radar as well as a weather station
and disdrometer. The second dataset was collected at the Jülich Observatory for Cloud Evolution Core Facility (JOYCE-CF,
Löhnert et al., 2015) which is located ca. 50 km north of Meckenheim. JOYCE-CF is regularly equipped with cloud radars as
well as a suite of remote sensing and in situ instruments including disdrometers. The permanently installed instrumentation
has been extended by additional cloud radars and disdrometers during the measurement campaign "TRIple frequency and
Polarimetric radar Experiment for improving process observation of winter precipitation" (TRIPEX-pol) which took place from
October 2018 until February 2019. The larger range of rainfall rates observed at the RPG site allows to test both calibration
methods with the same dataset. The continuous observations at JOYCE are lacking more intense rainfall rates (larger than
7 mm h$^{-1}$) required for the self-consistency method, but the longer time series allow for more detailed evaluation of the
calibration performance using disdrometers.

### 2.1 Radars

For this study, we use two 94 GHz cloud radars manufactured by Radiometer Physics GmbH (RPG), Meckenheim, Germany
(Fig. 1). The radars are based on solid state technology and use frequency modulated continuous wave (FMCW) signals. Note,
that the methods described in this study are also applicable to any other W-Band cloud radar (FMCW or pulsed) with a proper
rain mitigation system. An overview on the used radar design, operation, and the budget calibration was described in Küchler
et al. (2017). Typical radar specifications are summarized in Table 1. Configuration, maintenance, and observation periods for
each radar are given in Table 2. Throughout the paper, the radars are denoted according to their numbers in Table 2 (see first
column).





## 2.2 In situ instruments

The radars are equipped with Vaisala WXT520 weather stations (Basara et al., 2009) which provide atmospheric pressure, temperature, relative humidity, as well as one minute averaged rainfall rate derived from a piezoelectric sensor. The optical disdrometer PARSIVEL[2] (hereafter Parsivel, Löffler-Mang and Joss, 2000; Tokay et al., 2014) and the rain weighing gauge PLUVIO[2] (Pluvio throughout the paper) are manufactured by OTT Hydromet GmbH, Kempten, Germany. They belong to the permanently installed instrumentation of JOYCE (roof platform, 17 m above ground level). Due to a site maintenance, Parsivel and Pluvio were operated until 27 Nov 2018 at a nearby roof in ca. 50 m distance from the radars. From 27 Nov 2018 on, both instruments were installed back very close to the radars with distances of less than 10 m. The Pluvio installed at JOYCE has a 200 cm$^2$ orifice and a single Alter type wind shield (OTT Precipitation Wind Shield, Kochendorfer et al., 2017). Data are recorded with a one-minute averaging period; the real time output product is used for this study. Parsivel is an optical disdrometer which uses a laser band to detect size and fall velocity of precipitating particles (Löffler-Mang and Joss, 2000; Löffler-Mang and Blahak, 2001; Tokay et al., 2014). The Parsivel software groups the measured drop sizes and velocities into predefined $32 \times 32$ matrix. The size and velocity bins can be found in Angulo-Martínez et al. (2018). Rain rate and reflectivity are calculated using the raw data ($32 \times 32$ Matrix). A similar optical disdrometer, the Laser Precipitation Monitor (LPM, Fig. 2a) from Adolf Thies GmbH (Angulo-Martínez et al., 2018), is continuously operated at RPG site since 14 June 2018. The LPM collected data during summer 2018 at RPG site; from 1 Nov 2018 to 6 Dec 2018 the LPM was installed at the JOYCE site as part of TRIPEx-pol campaign. The LPM provides a particle-event mode where a message with the size and velocity of each individual particle is generated (Prata de Moraes Frasson et al., 2011). The particle-event mode is normally used for calibration purposes. The particle's size and velocity is provided separately assuming either a spherical or a "hamburger" shape. The later shape lacks a detailed description in the LPM manual, hence, we decided to only use the values for the spherical shape. Prata de Moraes Frasson et al. (2011) report that the data transfer rate may not be sufficient for a large number of particles. The manufacturer also notes in the LPM manual that not all particles might be registered at high precipitation rates. Unfortunately, a more detailed explanation of this issue and whether it is related to the data transfer rate or to other well-known issues of optical disdrometers, such as multiple particles in the field of view or partial beam filling, is missing. We developed a test device to estimate the underestimation of events due to limited data transfer rate. A chopper wheel with 2 closed and 2 open quadrants was mounted in a way to completely block or open the LPM laser beam (Fig. 2b). The event frequency was registered with a photo transducer and subsequently increased in steps from 3.7 to 83.2 s$^{-1}$. The data from the LPM was transferred using a serial RS485 full-duplex connection with 115 kBaud transfer rate. The LPM detected the event rate accurate with 1 s$^{-1}$ up to 77 s$^{-1}$. Larger event rates were significantly underestimated by the LPM. If we assume a Marshall-Palmer distribution, the event rate due to a rainfall rate of 20 mm h$^{-1}$ is 30 s$^{-1}$. As most rainfall events analyzed in this study are well below this rainfall rate, the LPM data transfer problem is unlikely to introduce large uncertainties. A more serious issue for rainfall measurements with the LPM are splashing effects which have been found by Angulo-Martínez et al. (2018) to cause up to 20% overestimation of the particle number. In order to reduce splashing effects, we covered all the LPM surfaces with spongy and cotton material (Fig. 2a). In order to further reduce the effects of splashing on the calculated rain rate and reflectivity, we





follow the approach of Tokay et al. (2014) and reject all particles with velocities deviating by more than a factor of 2 from a
theoretical size-velocity relation (Foote and Du Toit, 1969).

   Figure 3 shows a comparison of measured one-minute rain rates from the four in-situ sensors. The basis for the comparison
are observations from 1 Nov 2018 to 6 Dec 2018. In total we found 391 minutes of precipitation detected by all sensors. The
observed rain rates were mainly below 7 mm h$^{-1}$. The correlation between LPM and Parsivel rainfall rates is 0.96; LPM shows
slightly smaller values than Parsivel. The two weather stations show a correlation with disdrometers varying from 0.84 to 0.88.
These correlations are in an agreement with Prata de Moraes Frasson et al. (2011). The one-minute rainfall rates provided by
Pluvio were found to be very noisy with correlations to the other in situ sensors ranging from 0.5 to 0.6. Nevertheless, the one-
day accumulated precipitation from Pluvio correlates well with those from the Thies and Parsivel (0.997 and 0.99, respectively,
calculated with 10 rainy days). As Pluvio is a weighting gauge, it measures mass representing accumulation of droplets in the
bucket. The rainfall rate is derived as the time derivative of accumulated mass which can lead to more noisy rainfall rates. In
contrast, optical disdrometers measure every single droplet crossing the laser beam and calculate the accumulated rainfall as
integral over time. The accumulated precipitation from WXT520 weather stations has not been stored and therefore cannot be
analyzed.

## 3   Method 1: The self-consistency method for W-band polarimetric cloud radars

Goddard et al. (1994) developed a calibration approach for 3 GHz radars based on observations of reflectivity $Z$, differential
reflectivity $Z_{DR}$, and specific differential phase shift $K_{DP}$ in rain at low elevation angles. At S-band, $Z_{DR}$ defines the $K_{DP}/Z$
ratio because $Z$ and $K_{DP}$ depend on the number concentration of droplets, while $Z_{DR}$ is a proxy for drop median size
(Ryzhkov et al., 2005; Kumjian, 2013). For $Z_{DR}$ exceeding 2 dB, which is often observed in strong rainfall, the relation
between the three parameters is not affected by DSD variability. $Z_{DR}$ and $Z$ profiles can thus be used in strong rainfall to
reconstruct the expected differential phase $\Phi_{DP}$ profile. The radar is considered to be well calibrated if the expected and the
measured profiles of $\Phi_{DP}$ agree. According to Chandrasekar et al. (2015), a standard accuracy, which can be achieved for
$Z_{DR}$, is about 0.1 dB. $K_{DP}$ is calculated as a range derivative and, therefore, is immune to the radar polarimetric calibration.
As a small bias in $Z_{DR}$ affects the expected $\Phi_{DP}$ profiles much less than a bias in $Z$, any difference between measured and
expected profiles of $\Phi_{DP}$ is assigned to a reflectivity offset. The reflectivity calibration factor is then simply determined by
shifting the reflectivity profile until a minimum between the estimated and measured profiles of $\Phi_{DP}$ is reached.
Hogan et al. (2003) noticed that the method of Goddard et al. (1994) is not directly applicable to W-band radars for the
following reasons. First, attenuation due to rain is almost negligible at 3 GHz while it strongly increases towards higher fre-
quencies. Second, non-Rayleigh scattering causes reflectivity at W-band to increase much less with rainfall rate as compared
to lower frequencies. As a result, W-band reflectivities become less sensitive to rain rate for increasing rain intensities (Hogan
et al., 2003). Third, in contrast to lower frequencies, $Z_{DR}$ at W-band does not exceed 0.12 dB for rain rates up to 150 mm h$^{-1}$
(Aydin and Lure, 1991). Fourth, estimation of $K_{DP}$ from radar observations becomes more complicated. Otto and Russchen-
berg (2011) and Trömel et al. (2013) show that the total measured phase shift is the sum of a backscattering and a propagational





component (see Eq. B6). At low frequencies, the backscattering phase shift $\delta$ is usually negligible but it increases with larger frequencies. At mm-wavelengths, even relatively small drops in the range of 2–3 mm diameter produce up to 10° backscattering differential phase (Matrosov et al., 1999). For a polarimetric calibration method applicable to mm-wavelengths it is thus

crucial to find a way how to separate $\delta$ and $K_{DP}$. In order to find a solution for the above-mentioned problems, we identify a set of different propagation and backscattering variables, to which an approach similar to Goddard et al. (1994) can be applied. To infer suitable relations between radar observables, we simulate them using the T-Matrix model (Mishchenko, 2000) and a range of PSDs similar to Hogan et al. (2003). We assume normalized gamma distributions with $\mu$ from 0 to 15 and $N_L$ from $5 \times 10^2$ to $2.5 \times 10^4$ mm$^{-1}$ m$^{-3}$. For given $\mu$ and $N_L$, the median volume diameter $D_0$ was increased in 0.05 mm steps starting

at 0.1 mm until the rain rate reached 20 mm h$^{-1}$. Detailed description of how the various radar variables ($Z_0$, $A$, $Z_{DR}$, $K_{DP}$, $A_{DP}$, and $\delta$) are calculated can be found in Appendices A and B.

### 3.1   Replacement for $Z_{DR}$

As discussed above, Mie scattering effects complicate the use of $Z_{DR}$ at W-Band and we need to find an alternative parameter which is closely related to $D_0$. Trömel et al. (2013) found at X-Band that $\delta$ is a suitable parameter which is independent of $N_L$

and sufficiently related to $D_0$. As can be seen in Figure 4, $\delta$ is nearly directly proportional to $D_0$ at W-Band for rain rates up to 7 mm h$^{-1}$ and even at larger rain rates, $\delta$ seems to be a reasonable proxy for $D_0$. Thus, we will use $\delta$ in the following as a replacement for $Z_{DR}$ used at lower frequencies in order to find relations between $Z_0$ and propagation variables.

### 3.2   Relations between propagation and backscattering variables

In the original method of Goddard et al. (1994), a ratio of the propagation parameter $K_{DP}$ and the backscattering parameter

$Z_0$ is parameterized as a function of $Z_{DR}$ characterizing the median drop size. Using the large set of rain PSDs and forward simulated radar parameters, we can parameterize the ratio $K_{DP}/Z_0$ as a function of $\delta$ for W-Band:

$$\frac{K_{DP}}{Z_0} = a_1 f(a_2\delta + a_3) + a_4 f(a_5\delta + a_6) + a_7 f(a_8\delta + a_9) + a_{10}, \tag{1}$$

At frequencies where rain attenuation is non-negligible, we also need to parameterize specific attenuation $A$. The backscattering differential phase shift $\delta$ defines also the ratio $A_{DP}/A$:

$$\frac{A_{DP}}{A} = b_1 f(b_2\delta + b_3) + b_4 f(b_5\delta + b_6) + b_7 f(b_8\delta + b_9) + b_{10}. \tag{2}$$

We also introduce an additional relation to constrain relations between $Z_0$ and $\delta$. This is done by coupling these two parameters via the absolute value of the specific attenuation $A$ in dB km$^{-1}$:

$$A = c_1 f(c_2\delta + c_3 Z_0 + c_4) + c_5 f(c_6\delta + c_7 Z_0 + c_8) + c_9 f(c_{10}\delta + c_{11} Z_0 + c_{12}) + c_{13} f(c_{14}\delta + c_{15} Z_0 + c_{16}) + c_{17}. \tag{3}$$

In Eqs. 1–3 $f$ is the following function:

$$f(x) = \frac{2}{1 + e^{-2x}} - 1. \tag{4}$$





The fit coefficients $a_{1-10}$, $b_{1-10}$, and $c_{1-13}$ are given in Tables A1, A2, and A3, respectively. In Eqs. 1–3 the units of $Z_0$, $A$, $K_{DP}$, $A_{DP}$, and $\delta$ are mm$^6$ m$^{-3}$, dB km$^{-1}$, $^\circ$ km$^{-1}$, dB km$^{-1}$, and $^\circ$ respectively. In the supplementary materials we provide Matlab/Octave functions for Eqs. 1–3.

Figure 5 shows the simulated polarimetric variables and the fitted approximations (Eqs. 1–3). The remaining RMSE of the $K_{DP}/Z_0$, $A_{DP}/A$, and $A$ approximations are $2.3 \times 10^{-4}$ $^\circ$km$^{-1}$m$^3$ mm$^{-6}$, $3.2 \times 10^{-4}$ dB km$^{-1}$ dB$^{-1}$ km, and 0.3 dB/km, respectively. Figure 5a indicates that at $\delta$ close to 0.5 $^\circ$ $K_{DP}/Z_0$ is close to 0, which represents a limit of the self-consistency method. The method becomes robust at $\delta$ values exceeding 1 $^\circ$.

### 3.3   Separating propagational and backscattering components using Doppler spectra

Profiles of $Z_0$, $A$, $\delta$, $A_{DP}$, and $K_{DP}$ are not directly measured by a dual-polarized cloud radar. Instead, the radar measures

variables ($Z$, $Z_{DR}$, and $\Phi$), which are combinations of propagational and backscattering effects as can be seen in Eqs. A1, B5 and B6. Several studies presented approaches to separate propagational and backscattering components for centimeter wavelength radars (Otto and Russchenberg, 2011; Schneebeli and Berne, 2012; Trömel et al., 2013). These approaches are based on relations between profiles of $Z_{DR}$ and $\delta$ (Otto and Russchenberg, 2011; Schneebeli and Berne, 2012), and $A$ and $K_{DP}$ (Trömel et al., 2013). However, as already discussed above, those methods cannot be applied to W-Band because of non-

Rayleigh scattering and attenuation effects. As a result, $Z_{DR}$ becomes less informative, and relations between $A$ and $K_{DP}$ vary for different DSD when $\delta$ exceeds 1$^\circ$.

    A common approach to separate backscattering from propagational effects is the use of Doppler spectra. In the absence of strong turbulence, smaller droplets populate in the slow falling part of the spectrum while the larger drops are found on the fast falling side. Due to the relatively well-known relation of drop size and terminal velocity, the spectral power at each velocity

bin can be associated to a certain drop size range (Kollias et al., 2002). The small droplets can be assumed to be only affected by propagational effects while the larger drops are also affected by Mie scattering effects. Therefore, the spectral information can be used to separate the two components in low turbulence conditions. This approach has been applied to non-polarimetric dual-wavelength spectra in rainfall and snow to separate attenuation and Mie scattering effects (Tridon and Battaglia, 2015; Tridon et al., 2017; Li and Moisseev, 2019). Here we follow the same idea but with polarimetric spectra.

Polarimetric Doppler spectra have only been sporadically used in the past, probably due to the demands regarding storage capacity and required high data quality. At centimeter wavelength, their potential has been shown for microphysical retrievals (Moisseev and Chandrasekar, 2007; Spek et al., 2008; Dufournet and Russchenberg, 2011; Pfitzenmaier et al., 2018) and efficient clutter suppression (Unal, 2009; Moisseev and Chandrasekar, 2009; Alku et al., 2015). The number of installed polarimetric Doppler cloud radars is only recently increasing with only a few studies so far exploring their potential for

microphysical studies and retrievals (Oue et al., 2015; Myagkov et al., 2015, 2016b; Oue et al., 2018).

    As shown by Aydin and Lure (1991), drops up to a size of 1.2 mm do not produce a strong backscattering differential reflectivity $z_{DR}$ at W-Band. At sizes larger than 1.2 mm, the $z_{DR}$ spectrum reveals a series of minima and maxima. The authors also simulated a velocity $z_{DR}$ spectrum for 1 mm h$^{-1}$. The values of $z_{DR}$ are nearly 0 dB below 3 m s$^{-1}$ terminal velocity and, therefore, any changes in $Z_{DR}$ in this terminal velocity range can be addressed to differential attenuation.



We therefore derive differential reflectivity from the Doppler velocity range 0–2 m s$^{-1}$, where we assume all particles to be Rayleigh scatterers (hereafter refereed as the small-size part of a Doppler spectrum). Estimating this small-particle $Z_{DR}$ individually for each range bin directly provides us with the profile of the cumulative differential attenuation $DA$:

$$DA(r) = C_{DA} - 2 \int_0^r A_{DP}(r) dr, \qquad (5)$$

where $C_{DA}$ is an offset in differential reflectivity in dB due to the polarimetric calibration. Uncertainties of the $DA$ profile can be characterized using the variances of $Z_{DR}$ over the small-size part of the spectra. Unfortunately, Aydin and Lure (1991) do not show the size spectrum of $\delta$. Nevertheless, as it is shown for lower frequencies (Matrosov et al., 1999; Ryzhkov, 2001; Trömel et al., 2013) and as we further show in Sec. 3.6, the spectrally resolved $\delta$ shows a similar oscillatory behaviour as $z_{DR}$. Applying the same approach as described above, we can estimate the cumulative differential phase shift $DP$ from the $\Phi_{DP}$ in the small-size part of the spectrum:

$$DP(r) = C_{DP} + 2 \int_0^r K_{DP}(r) dr, \qquad (6)$$

where $C_{DP}$ is an offset in differential phase in ° due to the polarimetric calibration. The profile of $\delta$ can be simply estimated by subtracting $DP$ from the $\Phi_{DP}$ profile (see Eq. B6).

### 3.4 Algorithm

The different modules of the method are illustrated in Fig. 6. The method is based on finding a state vector corresponding to an optimal match of expected and the observed radar variables. The matching is achieved by minimizing a cost function using a global stochastic optimization method called differential evolution (DE) approach (Storn and Price, 1997). DE was recently used by Rusli et al. (2017) for a detailed characterization of drizzle and cloud liquid. In this study, we use the built-in Octave implementation of DE, which is based on Das et al. (2009). We use the default strategy DEGL/SAW/bin with a mutation factor of 0.8, a crossover probability of 0.9, a tolerance of $10^{-3}$, maximum number of iterations of 200, and a population size NP= $20N_v$, where $N_v$ is the number of elements in the state vector. DE stops when the maximum number of iterations is reached or the relative difference in the cost function between the best and the worst state vector in the population is below the specified tolerance. When DE reaches one of the stopping criteria, the state vector with the lowest cost function is taken as the output.

The state vector contains a range profile of $A(r)$ [dB km$^{-1}$] and the calibration factors $C_Z$ [dB], $C_{DA}$ [dB], and $C_{DP}$ [°]. DE does not require an a priori state vector. Instead, it requires realistic limits for each element of the state vector (Table 3). Within each iteration, the DE algorithm stochastically creates NP state vectors.

From each generated state vector a profile of $Z_0$ is calculated as follows:

$$Z_0(r) = Z(r) + C_Z + 2 \int_0^r [A(r) + A_g(r)] \, dr - 10 \log_{10} |K_0|^2 + 10 \log_{10} |K|^2, \qquad (7)$$



where $Z(r)$ is the measured reflectivity profile in dBZ, $|K_0|^2$ is the dielectric factor assumed by the radar (0.74 for our radars).

Surface observations of temperature, relative humidity, and pressure are used to estimate the attenuation profile due to gases $A_g(r)$ (see Sec. A2). The dielectric factor $|K|^2$ is calculated for liquid water at surface temperature. Using profiles of $Z_0$ and $\delta$, an expected profile $DP'$ is found using Eq. 1. The prime is used to discriminate the expected variable from the one estimated from measurements. Profiles of $A$ and $\delta$ are used to estimate the expected $DA'$ profile from Eq. 2. Finally, the expected profile of $A'$ is calculated using the expected profiles of $DA'$ and $DP'$ and Eq. 3.

The profiles of $A'$, $DA'$, and $DP'$ are further used for the calculation of the cost function $CF$:

$$CF = CF_{DA} + CF_{DP} + CF_A, \tag{8}$$

where

$$CF_i = [\boldsymbol{w}_i - \boldsymbol{W}_i]^T \mathbf{S}_i^{-1} [\boldsymbol{w}_i - \boldsymbol{W}_i]. \tag{9}$$

In Eq. 9, $i$ specifies a variable and $\boldsymbol{w}_i$ contains the profile of the expected values for the i-th variable ($DA$, $DP$, and $A$). The vector $\boldsymbol{W}_i$ contains the profile of the i-th variable inferred from measurements. The attenuation profile in the current state vector is taken as $\boldsymbol{W}_A$. $\mathbf{S}_i$ is the error covariance matrix of the i-th variable. Non-diagonal elements of $\mathbf{S}_i$ are assumed to be 0 since no correlation between errors in different range bins is expected. Based on uncertainty estimates of $A$ (see Sec. 3.2) related to uncertainties in the approximation Eq. 3, diagonal elements of $\mathbf{S}_A$ are set to $(0.3 \text{ dB km}^{-1})^2$. Estimation of $DA$, and $DP$ from radar observations as well as their diagonal elements in the error covariance matrix is done as described in Sec. 3.3.

## 3.5 Uncertainties of the method

In order to estimate the uncertainties of the method, we simulated 1000 samples of slanted 1 km profiles of $Z_0$, $A$, $A_{DP}$, $K_{DP}$, and $\delta$, as described in Appendices A and B. For the simulations, the normalized gamma DSD were used with $\mu$ and $N_L$ randomly chosen for each sample and each range bin. The ranges of $\mu$ and $N_L$ were from 0 to 15 and from $5 \times 10^2$ to $2.5 \times 10^4 \text{ mm}^{-1} \text{ m}^{-3}$, respectively. Size distributions with $A$ less than 3 dB km$^{-1}$ were excluded from the analysis because such attenuation values are close to the magnitude of the measurement variability. In order to take into account measurement variability of radar reflectivity, a random Gaussian noise was added to $Z_0$ with variance set to $Z_0^2/20$, where 20 is typical number of spectra averaged by the used radars (Eq. 5.193 in Bringi and Chandrasekar, 2001). Taking into account that signal-to-noise ratio in rain within the first kilometer typically exceeds 30 dB, variability in the polarimetric variables are low (Sec. 6.5 in Bringi and Chandrasekar, 2001) and are thus neglected. With the simulated variables the profiles of $Z$, $DA$, and $DP$ were derived. Variability in calibration constants $C_Z$, $C_{DA}$, and $C_{DP}$ were randomly generated assuming uniform distributions given in Table 3 and added to $Z$, $DA$, and $DP$, respectively. The uncertainties in $C_Z$, $C_{DA}$, and $C_{DP}$ are assumed to be the same for all range bins for a single sample. In order to take into account uncertainties in the separation of $\delta$ and $DP$, we added a random Gaussian noise with standard deviation of 0.5° to $DP$ and subtracted the corresponding values from $\delta$. Similarly, a random Gaussian noise with standard deviation of 0.3 dB was added to $DA$. All Noise values were different for each range bin and each sample.





The method was tested using the simulated profiles of $Z$, $DA$, and $DP$ as input. For each sample, the best estimate of $C_Z$ provided by the algorithm was then compared to $C_Z$ used for the simulation. The results shown in Fig. 7 show that 90% of the differences are within $\pm 0.7$ dB.

### 3.6 Application to measurements from radar 2

We now exemplarily demonstrate the different steps of the self-consistency method with a case study. A precipitation event, which includes drizzle and stronger rainfall was observed at Meckenheim on 9th June 2018 operating the radar at $30°$ elevation (Fig. 8). The melting layer can be depicted at 2.5 km by enhanced values of $Z_{DR}$ and $\Phi$. During the period between 18 and 21 UTC, the rain sensor only registered drizzle on the ground, while later, a short and more intense rainfall event with up to 15 mm h$^{-1}$ rainfall took place. As expected, $Z_{DR}$ and $\Phi$ are close to zero in the drizzle part due to the near spherical

shape of the drops. Non-zero values are found during the stronger rainfall event due to the larger and hence more aspherical raindrops. Around 21 UTC positive and negative values in both $Z_{DR}$ and $\Phi_{DP}$ are visible. These values indicate presence of backscattering and propagational effects of rain drops.

Backscattering and propagational effects can be better separated when moving to the Doppler spectral space (Fig. 9). The spectra during the stronger rainfall event show the expected oscillatory behaviour for larger Doppler velocities which are

principally related to larger sizes. It should be noted that we did only apply a very rough correction for horizontal wind as the method itself is not dependent on such a correction. The main goal in the spectral analysis is the separation of the Rayleigh scattering part (only affected by propagational effects) from the Mie scattering part (affected by both backscattering and propagational effects). The spectral part which is not affected by oscillations (approximately Doppler velocities slower than $-2$ m s$^{-1}$) shows decreasing values for $Z_{DR}$ and $\Phi$ with increasing range caused by propagational effects (see also spectra

plotted for constant ranges in Fig. 10). The reflectivity spectra themselves show somewhat unexpected increase with range and also the oscillations are less pronounced than in the polarimetric variables. This can be explained by the fact that $Z$ is also dependent on the particle concentration.

The spectral regions with oscillatory behaviour represent drop sizes for which backscattering and propagational effects are convoluted (Fig. 10). In the part where the smaller droplets scatter in the Rayleigh regime, we find a plateau-like region in

the spectra of $Z_{DR}$ and $\Phi_{DP}$ (Fig. 10). The deviation of the plateau from 0 dB indicates the propagational effects. It should be noted that calibration offsets would of course also result in a shifted plateau region, however, independent of range. $Z_{DR}$ and $\Phi$ of radar 2 have been calibrated using zenith observations in light rain as described in Myagkov et al. (2016a). Vertical observations in light rain show $Z_{DR}$ and $\Phi_{DP}$ values of $1.003 \pm 0.01$ (linear units) and $0 \pm 0.15°$. After proper calibration, we can assign the shift of the plateau region solely to propagation effects for which we derive mean and standard deviation for each

range (Fig. 11). $A_{DP}$ and $K_{DP}$ are found for this case to be on average about 0.13 dB km$^{-1}$ and -0.95$°$ km$^{-1}$, respectively.

In order to estimate a profile of $\delta$, which is used as an input for Eqs. 1 and 2, the profile of $DP$ shown in Fig. 11b is subtracted from the profile of $\Phi_{DP}$. Profiles of $\Phi_{DP}$ and $\delta$ are shown in Fig. 11c.





Figure 12a and b show the best fits for $DA$ and $DP$ profiles found by the optimization algorithm. The resulting best matching radar calibration coefficient for reflectivity $C_Z$ has been found for this time sample to be –0.7 dB meaning that the radar 2 is

slightly underestimating reflectivity values.

The self consistency method allows for an evaluation of the radar calibration even from a single sample. In order to test how repeatable the results of the self-consistency method are, we applied the method to 64 samples from 3 rain events. For the method to produce reasonable results, the rainfall events and associated profiles have to fulfill the following criteria: (1) rain rate observed at the surface by the weather station must be below 20 mm h$^{-1}$, (2) $\delta$ has to be larger than 1° over at least 900 m

one-way range, (3) the median $K_{DP}$ must be lower than –0.3 ° km$^{-1}$, and (4) the median $A_{DP}$ is lower than –0.06 dB km$^{-1}$. The results shown in Fig. 13 indicate that the radar 2 has a small negative reflectivity bias. The mean $C_Z$ estimated from the 64 samples is –0.6 dB. The single-sample estimate of $C_Z$ from the 3 rain events varies from –1 to 0 dB which is within the uncertainty of the method estimated in Sec. 3.5. One might see an increasing trend in $C_Z$, but taking into account the method uncertainty of $\pm0.7$ dB and the few cases, the trend is not statistically significant. As it will be shown further in the next section,

certain variability in $C_Z$ can be explained by imperfections in the removal of liquid water from the radome.

## 4   Method 2: Disdrometer-based method

The second method is based on a comparison of measured reflectivies (denoted as $Z_m$) at distances close to the surface with calculated reflectivities based on DSDs observed by collocated disdrometer ($Z_d$ hereafter). Values of $Z_d$ are calculated according to Appendix. A. This well-known approach is generally applicable to radars operating at any frequency, however,

the issue of variable rain properties between the lowest range gate and the disdrometer location remains a source of uncertainty. Using radar observations at the lowest range gates also requires that there are no antenna near field or receiver saturation effects and that wet radome or wet antenna effects are minimized. For a two-antenna system, such as used for the FMCW systems in this study, the incomplete beam overlap is corrected for using the method in Sekelsky and Clothiaux (2002). At ranges larger than 250 m from the radar, the beam overlap for all radars used is better than 90%. For the following analysis, we use $Z_m$ at

250 m range. It should be noted that the calculations in this section use altitude and not range; for slanted path, a conversion of range to altitude has to be applied. In the following, we will describe our approach how to mitigate two main sources of uncertainty for this method in the rainfall cases analyzed. A schematic of the entire processing chain is illustrated in Fig. 14.

### 4.1   Mitigating the effect of rain evaporation

Evaporation of rain on its way towards the surface is often observed at our sites. Figure 15 shows a simulation of the impact of

evaporation on the reflectivity at 250 m. It can be seen that in sub-saturated conditions the difference in radar reflectivity caused by evaporation can be strong. The effect is particularly pronounced for light precipitation where the difference can exceed 2 dB. In this case, the scattering is dominated by relatively small drops, whose diameters decrease faster due to evaporation than for big drops. In order to mitigate the effect of evaporation, we use an evaporation model described in Appendix A3. Based on temperature, relative humidity, and drop size at surface level, the model predicts the corresponding drop size at 250 m altitude.





For the calculations of drop sizes at 250 m altitude, all drops detected by the LPM within one minute prior to a radar sample time are used. In the case of Parsivel, which typically has a time resolution of 1 min, the data closest to a radar sample time are taken. The LPM in the single-event mode provides diameters of single particles and the evaporation model (Eq. A6) is directly applied to all detected drops. For Parsivel data, the evaporation model is applied to mean bin sizes and the number of particles per size bin is assumed to be constant with altitude. Note that in this case the width of the size bin changes with altitude. This

is equivalent to keeping the $32 \times 32$ raw data matrix - which is the standard output of Parsivel - constant but changing the mean drop diameters assigned to each matrix cell.

The estimated DSD for 250 m are then used for the calculation of $Z_0$ and $A$ at 250 m according to Table A1. Since the calculation of the whole attenuation profile with evaporation effect accounted for is time consuming, $A(r)$ is assumed to be constant and equal to the mean of $A$ values at the surface and at 250 m taken in dB km$^{-1}$.

## 4.2   Comparison of expected and measured reflectivity time series

In order to identify a potential time lag between $Z_m$ and $Z_d$, we calculate their temporal correlation assuming a range of time shifts. The time lag for which the maximum correlation is found is used for correcting the time series and the difference between $Z_m$ and $Z_d$ is analyzed. We recommend to only use $Z_m$ and $Z_d$ larger than 5 dBZ because the number of drops sampled by the disdrometer might be too low and not representative for smaller reflectivities.

## 4.3   Case study

We apply the disdrometer-based method to observations of radar 1 from 1 November 2018. From 12:30 to 16:20 UTC there was light precipitation at the JOYCE site. The mean precipitation rate was 0.5 mm h$^{-1}$ with maximum at 4.4 mm h$^{-1}$. The LPM operated in the particle-event mode (see Sec. 2.2). $Z_d$ are calculated according to Table A1. For calculating $Z_d$, all drop sizes have been corrected for evaporation. Figure. 16a shows the correlations between $Z_d$ and $Z_m$ at different time lags. The time

shift corresponding to the maximum correlation of 0.93 is $-65$ s which is applied to $Z_d$. After correcting for the time lag, $Z_m$ and $Z_d$ show a high correlation for values exceeding 5 dBZ (Figure 16b). A direct comparison of $Z_m$ and $Z_d$ (Figs. 16 c) shows that the mean difference in $Z_d - Z_m$ is –1.2 dBZ with a standard deviation of 0.3 dBZ (calculated according to Appendix. C).

## 4.4   Repeatability

In order to check how repeatable the results of the disdrometer-based method are, it was applied to LPM, Parsivel, and radar 1

observations in 12 rain events collected during the TRIPEX-pol campaign from 1 Nov 2018 to 6 Dec 2018. The same rain events are shown in Fig. 3. In the supplementary materials we provide figures similar to Fig. 16 and statistical analysis for each rain event.

Figure 17 shows the reflectivity differences $Z_d - Z_m$ for radar 1. The blue dots were calculated according to Sec. 4, while red dots are calculated without taking evaporation into account. Evaporation leads on average to about 0.7 dB underestimation

in $Z_d$, which might be critical if a reflectivity accuracy within $\pm 1$ dB is desired.


In Fig. 17 the differences $Z_d - Z_m$ are shown as functions of maximum rain rate observed by the disdrometers. At rain rates lower than 4 mm h$^{-1}$, values of $Z_d - Z_m$ vary from –2 to –0.9 dB and from –2.1 to –0.5 dB with mean values of –1.4 and –1.1 dB based on LPM and Parsivel (blue dots in Figure 17a and c), respectively. On average, reflectivity values based on Parsivel observations are about 0.3 dB larger than those from LPM. The reason for this difference are likely related to the

specific differences of the disdrometers, however, a detailed analysis of such differences is out of the scope of this study. A comprehensive comparison of LPM and Parsivel disdrometers can be found for example in Angulo-Martínez et al. (2018) and Johannsen et al. (2020).

The results for both disdrometers indicate a dependence of the calibration offset on maximum rain rate observed during a precipitation event. Since the slope of the linear regressions are similar with and without the evaporation correction, we

conclude that this effect may come from limitations of the rain mitigation system. However, the radome attenuation effects would certainly be much larger without a mitigation system. as shown by Hogan et al. (2003), those effects can easily exceeding 10 dB for strong rainfall.

## 4.5    Comparison with the self-consistency method

As has been shown in the previous section, the comparison of radar 1 observations with LPM and Parsivel shows that the radar

on average overestimates the reflectivity by 1.4 and 1.1 dB, respectively. The calibration of the radar 2 was estimated using the self-consistency methods and indicates that the radar underestimates the reflectivity by 0.7 dB.

During the TRIPEX-pol campaign the radar 1 and radar 2 were operating at the JOYCE site. The radar 2 was performing RHI and PPI scans. As it was mentioned in Sec. 4.5, the disdrometer-based method does not often show consistent results when applied to scanning data. Nevertheless, the radar 2 performed a PPI scan at the 85° elevation every 15 min. Therefore,

we used vertical observations from the radar 1 and the PPI scans from the radar 2 to find a reflectivity difference between the radars 1 and 2. The difference can be used to check the consistency of the two calibration evaluation methods. During the 12 rain events we identified more than 8000 samples for the comparison. For each sample of the radar 2, a closest time sample of the radar 1 was found. Within each sample we identified the closest range bins with reflectivity values exceeding 5 dBZ. The radar 1 shows on average 2.1 dB higher reflectivity values in comparison to the radar 2. This value is consistent with the

difference of $2.0 \pm 1.3$ dB between the biases found by the two methods separately for the radars 1 and 2.

## 5    Summary

Monitoring and evaluation of radar reflectivity calibration is a key requirement in order to provide long-term observational radar datasets to the cloud and precipitation community. In this study, we describe and compare two methods requiring very different degree of complexity in terms of instrumentation and retrieval technique. Both methods use natural stratiform rainfall

as reference target.

The first method is an extension of the widely used approach from Goddard et al. (1994) applied to precipitation radars. The original method uses the $Z_{DR} - K_{DP} - Z$ relation, but due to the more complex attenuation and scattering behaviour,





this method is not directly applicable to millimeter wavelengths. In this study, we provide a solution for this problems using spectral polarimetry obtained form a W-Band radar. The use of the spectral information allows to disentangle propagational and

backscattering effects. The method requires a observations of $Z_{DR}$ and $Phi_{DP}$ and is only applicable to slanted observations in rain with distinct backscattering and propagational polarimetric signatures. The backscattering phase shift should mostly exceed $1°$. The differential attenuation and specific differential phase shift should preferably be at least 0.05 dB km$^{-1}$ and 0.3° km$^{-1}$. The main advantage of this method is its low sensitivity to variabilities in rain DSD. We estimated the uncertainty of this method based on realistical assumtions of errors in the profiles of radar observables to be within $\pm 0.7$ dB.

We also tested and extended a much simpler and commonly used method, which compares reflectivities based on disdrometer DSDs measured at the surface with radar reflectivity at a close range gate. We included corrections for the time lag between the surface and elevated observation, as well as a correction for evaporation. The method allows for a repeatable evaluation of the radar reflectivity calibration within $\pm 0.9$ dB even with only a few hours of observations in rain with intensity below 4 mm h$^{-1}$. Averaging over a larger number of rain events allows to further reduce the uncertainties of the method. The disdrometed-based

method was tested with two common disdrometers of type LPM and Parsivel. The results do not differ by more than 0.4 dB.

The two methods were used to evaluate the reflectivity calibration of two W-Band radars. The self-consistency method showed that the radar 2 underestimates the reflectivity by about $0.7 \pm 0.7$ dB, while the disdrometer-based method indicated that the radar 1 overestimates the reflectivity by 0.5–2.1 dB. Unfortunately, the rainfall rates during the parallel operation of the two radars at JOYCE were not strong enough to compare the two methods directly. However, in case both methods provide

reliable estimates for each radar, the reflectivity difference seen by both radars should be close to the sum of both offsets. Indeed, the observed reflectivity difference is with 2.1 dB quite consistent with the difference of $2.0 \pm 1.3$ dB between the biases found by the two methods separately for the radars 1 and 2. We would also like to emphasize that a further evaluation of the two methods described here and other methods, e.g. using point target calibration or multi-frequency approach (Tridon et al., 2017) would be beneficial.

As wet radomes of W-band radar can cause attenuation exceeding 10 dB, the evaluation methods rely on efficient rain mitigation. We have found an evidence that the reflectivity bias of the used radars is correlated to the maximum rain rate, the slope of the linear regression is about 0.15 dB mm h$^{-1}$. The uncertainty of the used rain mitigation method is much smaller than uncertainties and wet antenna effects reported by (Hogan et al., 2003). Further investigations on this topic are required in order to understand effects limiting the performance of the used rain mitigation systems.

Summarizing the results, we recommend the disdrometer-based method for continuous monitoring of cloud radar calibration. Many operational sites are already equipped with disdrometers, which allows for a straight-forward application of the technique. The method can be applied to both vertical and slanted observations, though continuous scanning may limit the applicability of the method. The extended consistency-method is less sensitive to DSD variability and also allows calibration evaluation if only a few rainfall cases are available. Both methods can analogously be derived for Ka-Band systems.

*Code availability.* MATLAB/Octave functions for the approxmations Eqs. 1–3, and Eq. A6 are given in the suplementary material.





## Appendix A: Reflectivity calculation

The radar reflectivity factor $Z$ along a path of rainfall with constant properties can be calculated as

$$Z(r) = Z_0(r) - 2 \int_0^r \left( A(r) + A_g(r) \right) dr, \tag{A1}$$

where $Z_0$ [dBZ] is the non-attenuated reflectivity, $A$ [dB km$^{-1}$] is the one-way attenuation by rain, and $A_g$ [dB km$^{-1}$] is
one-way gas attenuation, $r$ is in km. The factor of 2 is related to the two-way propagation.

### A1    Non-attenuated reflectivity and attenuation by rain

For the DSD of rain we assume the widely used normalized gamma distribution (Illingworth and Blackman, 2002):

$$n(D) = \frac{0.033 N_L D_0^4 \Lambda^{\mu+4}}{\Gamma(\mu+4)} D^\mu \exp(-\Lambda D), \tag{A2}$$

where $\Lambda = (3.67 + \mu)/D_0$, $D$ is the equivalent sphere diameter in mm, $D_0$ is the median volume diameter in mm, $N_L$ is the
concentration parameter in mm$^{-1}$ m$^{-3}$, and $\mu$ is the distribution shape parameter. For numerical calculations, we discretized
the DSD with $D_i$ describing the mean diameter and $N_i$ denoting the drop number of the i-th size bin. $D_i$ are equispaced from
$10^{-2}$ up to 8 mm with a constant bin width of $10^{-2}$ mm.

     We approximate raindrops with oblate spheroids. The shape-size relation was taken from Pruppacher and Pitter (1971).
Based on video disdrometer observations, Huang et al. (2008) showed that raindrops are mostly aligned horizontally with
canting angle standard deviation of 8°. Aydin and Lure (1991) showed that this fluttering of drop orientation has a relatively
small effect at 94 GHz. Even for the reflectivity difference between horizontal and vertical polarization, for which one expects
the effect of particle orientation to be maximum, the differences do not exceed 0.12 dB for rainfall rates up to 150 mm h$^{-1}$. In
the following calculations we therefore assume the rain drops to be horizontally aligned.

     We calculate backscattering $S_{jk}(D_i)$ and forward $F_{jk}(D_i)$ scattering coefficients using the T-matrix method (Mishchenko,
2000). Here indices $j$ and $k$ stand for the polarization of transmitted and received waves, respectively. The temperature de-
pendence of the refractive index of liquid water is taken from Ray (1972). Using $S_{hh}(D_i)$ and $F_{hh}(D_i)$, the non-attenuated
reflectivity $Z_i$ in mm$^6$ m$^{-3}$ and specific one-way attenuation due to liquid water $A_i$ in dB km$^{-1}$ for one drop with the diameter
$D_i$ per unit volume are calculated with:

$$Z_i = \frac{10^{18} \lambda^4}{\pi^5 |K|^2} \left( 4\pi \left| S_{hh}(D_i) \right|^2 \right), \tag{A3}$$


$$A_i = 8.686 \times 10^3 \left( \frac{2\pi}{k} \mathrm{Im} \left[ S_{hh}(D_i) \right] \right), \tag{A4}$$

where $|K|^2$ is the dielectric factor of liquid water, $\lambda$ is the wavelength, and $k$ is the wave number.

     The final rainfall rate, reflectivity, and one-way attenuation are calculated as sum over the DSD. The equations used for the
different in situ instruments are summarized in Table A1.





**Table A1.** Summary of calculation formulas for rain rate, reflectivity, and attenuation for the normalized gamma DSD, DSD from Parsivel, and drops detected by LPM. For the normalized gamma DSD and Parsivel the index $i$ ranges over $n$ diameter bins, while for LPM it varies over $n$ drops detected by the instrument 1 minute prior to a radar sample time. The index $j$ moves over $m$ velocity bins of Parsivel. $C_{i,j}$ is the cell of the Parsivel raw data matrix. $V_i$ and $v_i$ are volume and terminal velocity of a drop with the diameter $D_i$, respectively. $Z_i$ and $A_i$ are reflectivity and attenuation for one drop with the diameter $D_i$ in a unit volume, respectively. $S_i = L_b(W_b - D_i/2)$ is effective sampling area of a disdrometer with $L_b$ and $W_b$ being the length and the width of the disdrometer laser beam (Tokay et al., 2014). $|K|^2$ is the dielectric factor of water at a certain temperature. $|K_0|^2 = 0.74$ is the constant dielectric factor set in the processing routine of the used radars.

| Parameter | Normalized gamma DSD | Parisvel | LPM |
|---|---|---|---|
| $R$ [mm h$^{-1}$] | $3.6 \times 10^6 \sum\limits_{i=1}^{n} N_i V_i v_i$ | $6 \times 10^4 \sum\limits_{i=1}^{n} \sum\limits_{j=1}^{m} \frac{C_{i,j} V_i}{S_i}$ | $6 \times 10^4 \sum\limits_{i=1}^{n} \frac{V_i}{S_i}$ |
| $Z$ [dBZ] | $10\log\left(\frac{|K|^2}{|K_0|^2} \sum\limits_{i=1}^{n} N_i Z_i\right)$ | $10\log\left(\frac{1}{60} \frac{|K|^2}{|K_0|^2} \sum\limits_{i=1}^{n} \sum\limits_{j=1}^{m} \frac{C_{i,j} Z_i}{v_i S_i}\right)$ | $10\log\left(\frac{1}{60} \frac{|K|^2}{|K_0|^2} \sum\limits_{i=1}^{n} \frac{Z_i}{v_i S_i}\right)$ |
| $A$ [dB km$^{-1}$] | $\sum\limits_{i=1}^{n} N_i A_i$ | $\frac{1}{60} \sum\limits_{i=1}^{n} \sum\limits_{j=1}^{m} \frac{C_{i,j} A_i}{v_i S_i}$ | $\frac{1}{60} \sum\limits_{i=1}^{n} \frac{A_i}{v_i S_i}$ |

## A2 Gas attenuation

Unlike for longer wavelength radars (e.g. precipitation radars), gas attenuation cannot be neglected for W-band. The major contributions to gas attenuation at W-Band are due to water vapor and oxygen which we calculate with the model by Liebe (1989). As continuous profile information of temperature, water vapor, and pressure are unavailable at RPG site and JOYCE-CF, we use the surface measurements of the weather station to approximate the vertical profiles. For the temperature profile, we use a constant empirical laps rate $K_t$. Based on the radiosonde database from Essen (Station number: 10410, 90 km from Meckenheim, 75 km from Jülich) the laps rate $K_t$ was estimated to be $4.8 \times 10^{-3}$ K m$^{-1}$. The launches from 1 Jan 2010 to 23 Oct 2018 with surface relative humidity exceeding 65% were used. For the $K_t$ estimation only the lowest 3 km of radiosonde ascends were taken. Relative humidity is assumed to be constant with height. The statistics of the calculated one-way gas attenuation profiles are shown in Fig. A1.

## A3 Drop evaporation

Xie et al. (2016) showed that change of the drop size due to evaporation can be derived from the following equation:

$$vD\frac{dD}{dH} = 4\frac{S-1}{F_K + F_D}, \tag{A5}$$

where $D$ and $v$ are the diameter and velocity of a drop, respectively, $H$ is a vertical range traveled by the drop, $S-1$ is the supersaturation with respect to liquid water, $F_K$ and $F_D$ are coefficients related to heat conduction and vapor diffusion, respectively. The calculation of $F_K$ and $F_D$ is based on Kumjian and Ryzhkov (2010).

Equation A5 relates an initial drop size at a certain altitude to the drop size at the surface. The disdrometer-based method requires an opposite relation, i.e. what would be the drop size at 250 m altitude if its size at the surface is known. The relation





can also be found by solving Eq. A5. The equation is solved numerically for surface diameters $D_s$ from 0.06 to 3 mm with a grid of 0.01 mm using an iterative approach. Large drops are less influenced by evaporation (Xie et al., 2016), therefore, for drops

larger than 3 mm the size change is neglected. The surface drop size is taken as the first guess of the drop size at 250 m $D_{250}$. Using Eq. A5 the corresponding size at the surface $D_{sm}$ is calculated and compared with $D_s$. In case the difference is smaller than 0.01 mm, $D_{250}$ is taken as the solution for corresponding $D_s$. If the difference is larger, $D_{250}$ is changed until the condition is satisfied. For minimization of the difference, the differential evolution method (Das et al., 2009) is applied, although any other optimization algorithm can be also used. Even though, the convergence for a single size is fast, the application of such

the evaporation correction to a number of sizes and different environment conditions is time consuming. Therefore, a set of precalculated $D_s$–$D_{250}$ relations at surface temperatures from 0 to 20°C with the 5°C step and surface relative humidity from 60 to 100% with the 5% step at the 1000 hPa surface pressure is used for the $D_s$–$D_{250}$ function approximation:

$$D_{250}(D_s, T, RH) = \begin{cases} \left[ \sum_{i=1}^{10} g_i f(p_i D_s + q_i T + u_i RH + \alpha_i) \right] + \beta, & \text{if } 0 < D_s < 3 \text{ mm} \\ D_s, & \text{if } D_s \geq 3 \text{ mm} \end{cases} \quad (A6)$$

where $D_{250}$ and $D_s$ are in mm, $T$ is in °C and has to be in the range from 0 to 30 °C, and RH is in % and should be in

the range from 60 to 100%. The coefficients $g$, $p$, $q$, $u$, and $\alpha$ are given in Table A4, $\beta = 20.038$. For the given ranges of the input parameters the root mean square difference between the simulated and approximated values of $D_{250}$ is 5.8 μm. In the supplementary materials we provide a Matlab/Octave function for Eq. A6.

## Appendix B: Polarimetric variables

The T-matrix calculations are also used to derive polarimetric variables such as backscattering differential reflectivity $z_{DR}$ [dB],

backscattering differential phase $\delta$ [°], one-way differential attenuation $A_{DP}$ [dB km$^{-1}$], and one-way propagation phase shift $K_{DP}$ [° km$^{-1}$]:

$$z_{DR} = 10 \log_{10} \left( \frac{\sum_{i=1}^{n} |S_{hh}(D_i)|^2 N_i}{\sum_{i=1}^{n} |S_{vv}(D_i)|^2 N_i} \right), \quad (B1)$$

$$\delta = \frac{180}{\pi} \arg \left[ \sum_{i=1}^{n} N_i S_{hh}(D_i) S_{vv}^*(D_i) \right], \quad (B2)$$

$$A_{DP} = 8.686 \times 10^3 \frac{2\pi}{k} \sum_{i=1}^{n} \text{Im} \left[ S_{hh}(D_i) - S_{vv}(D_i) \right] N_i, \quad (B3)$$

$$K_{DP} = \frac{180}{\pi} \frac{2\pi}{k} \sum_{i=1}^{n} \text{Re} \left[ S_{hh}(D_i) - S_{vv}(D_i) \right] N_i, \quad (B4)$$





where $*$ denotes the complex conjugation.

Differential reflectivity $Z_{DR}(r)$ [dB] and differential phase shift $\Phi_{DP}(r)$ [°] at a certain range $r$ [km] from the radar are a

sum of corresponding backscattering and propagational components:

$$Z_{DR}(r) = z_{DR}(r) + 2 \underbrace{\int_0^r -A_{DP}(r) dr}_{DA(r)},$$
(B5)

$$\Phi_{DP}(r) = \delta(r) + 2 \underbrace{\int_0^r K_{DP}(r) dr}_{DP(r)},$$
(B6)

where $DA$ and $DP$ are propagation components in differential reflectivity and differential phase shift, respectively.

**Appendix C: Variance of a numerical average**

A variance (denoted as var) of an average of $N_s$ samples can be found as follows:

$$\text{var}\left(\frac{1}{N_s} \sum_{i=0}^{N_s-1} s_i\right) = \frac{1}{N_s^2} \sum_{i=0}^{N_s-1} \sum_{j=0}^{N_s-1} \text{cov}(s_i, s_j),$$
(C1)

where cov stands for covariance, $s$ is a sample with a lag is indicated by the subscripts $i$ and $j$. The covariance $\text{cov}(s_i, s_j)$ is calculated as a multiplication of the standard deviations of the corresponding variables and their correlation. Assuming that the

analyzed process is stationary with the standard deviation $\sigma_s$, $\text{cov}(s_i, s_j)$ can be written as follows:

$$\text{cov}(s_i, s_j) = \rho_\tau \sigma_s^2.$$
(C2)

Here $\rho_\tau$ is the normalized auto-covariance function at the lag $\tau = i - j$. Substituting Eq. C2 into Eq. C1:

$$\text{var}\left(\frac{1}{N_s} \sum_{i=0}^{N_s-1} s_i\right) = \frac{\sigma_s^2}{N_s^2} \sum_{i=0}^{N_s-1} \sum_{j=0}^{N_s-1} \rho_\tau.$$
(C3)

Similar relation for analytic functions was derived by Leith (1973). In the case of uncorrelated samples, the normalized auto-

covariance function is a delta function, the double sum in Eq. C3 is equal to $N_s$, and the variance can be found using the well-known relation $\sigma_s^2/N_s$. This relation is widely used in the weather radar community for improving the signal detection (Eq. 5.193 in Bringi and Chandrasekar, 2001; Görsdorf et al., 2015). In contrast, when all the samples are highly correlated within the averaging period, the double sum is equal to $N_s^2$ and, as expected, the variance of the average does not change. In the general case, when the analyzed process has a certain coherency time, the variance is within the range between $\sigma_s^2/N_s$ and

$\sigma_s^2$.





*Author contributions.* AM developed and tested the self-consistency method. AM and SK added the time-lag and evaporation corrections to the disdrometer-based method. SK organized the TRIPEX-pol measurement campaign. AM applied the disdrometer-based method. AM and SK evaluated the results of the calibration evaluation. AM and SK prepared the manuscript. TR developed the scanning polarimetric W-band radar and reviewed the manuscript.

*Competing interests.* S. Kneifel has no competing interests. A. Myagkov and T. Rose are employees of Radiometer Physics GmbH.

*Acknowledgements.* The contributions by S. Kneifel have been funded by the Deutsche Forschungsgemeinschaft (DFG, German Research Foundation) under grant KN 1112/2-1 as part of the Emmy-Noether Group OPTIMIce. The TRIPEx-pol campaign has also been supported by the DFG Priority Program SPP2115 "Fusion of Radar Polarimetry and Numerical Atmospheric Modelling Towards an Improved Understanding of Cloud and Precipitation Processes" (PROM) under grant PROM-IMPRINT (Project Number 408011764). We would also like
to acknowledge Juan-Antonio Bravo Aranda, University of Granada, for providing the vertically-pointed LDR-mode W-Band radar for the TRIPEx-pol campaign. We also acknowledge the staff of the University of Cologne, research center Jülich, and RPG, especially B. Bohn, R. Haseneder-Lind, and A. Saljihi for their help with the installation of the W-band radars, and K. Schmidt for the preparation of the scanning polarimetric W-band radar for the TRIPEx-pol campaign.



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



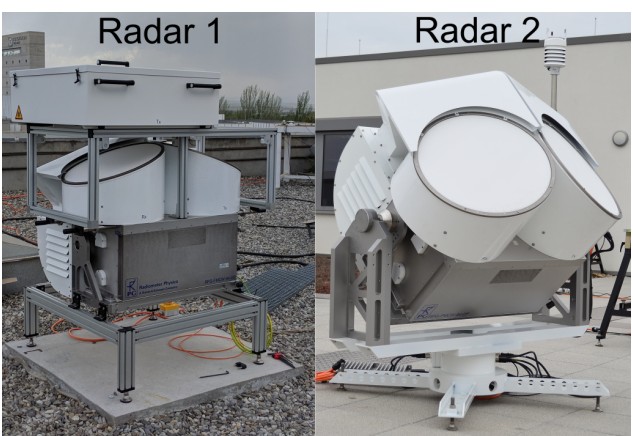

**Figure 1.** Impressions of the two FMCW W-Band radars used in this study as indicated in Table 2. Photos courtesy of the radar owners.



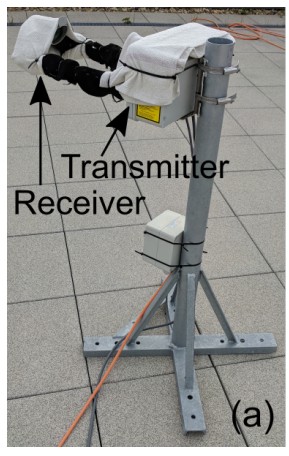

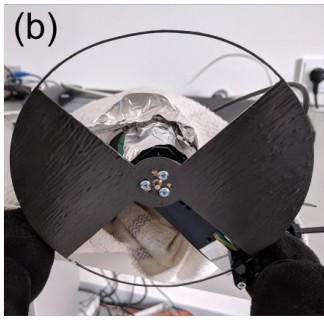

**Figure 2.** The Laser Precipitation Monitor (LPM) at the RPG site (a). Metal surfaces close to the laser beam are covered with spongy/cotton material for splashing mitigation. The chopper wheel (b) was mounted on side of the LPM detector for testing data transmission rate (see text for details).



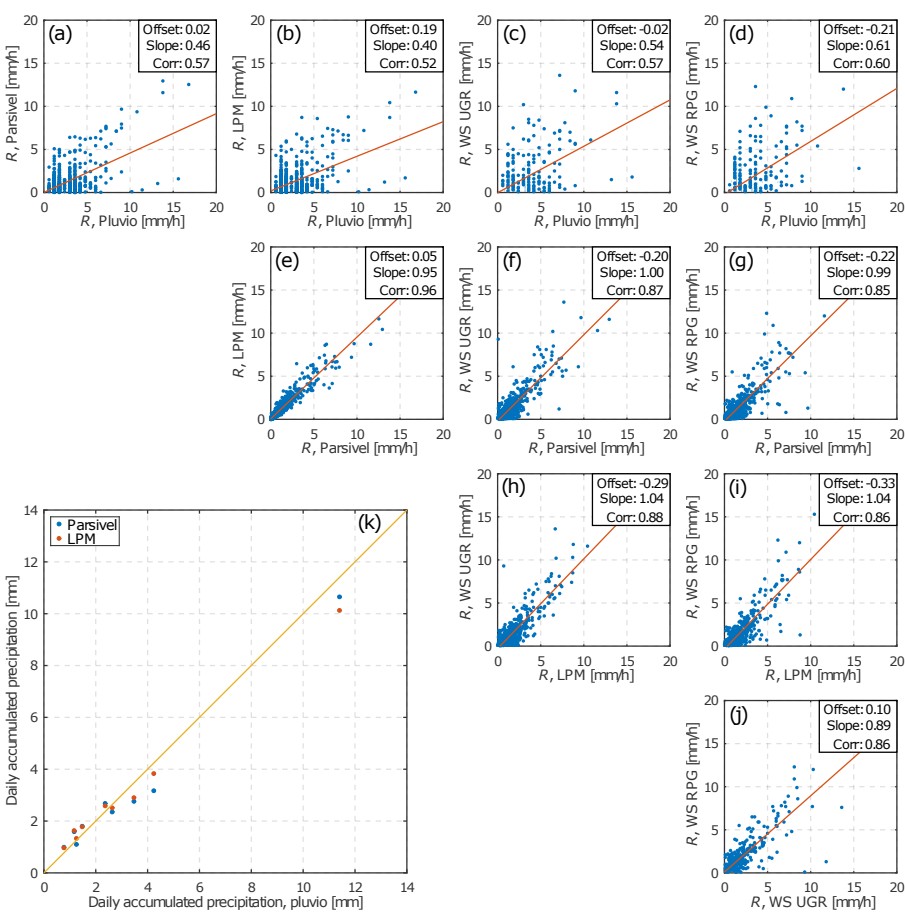

**Figure 3.** A comparison of one-minute rain rates observed by Pluvio, Parsivel, LPM, and the two WXT520 weather stations attached to the radars (a-j). The weather stations of radars 1 and 2 are denoted as WS UGR and WS RPG, respectively. The dataset from the TRIPEX-pol campaign (1 Nov 2018 to 6 Dec 2018) contains 391 minutes of rainfall detected by all sensors simultaneously. Each subplot contains estimated offset and slope of a linear fit (red line) as well as the Pierson correlation coefficient. Panel (k) shows a comparison of the daily accumulated precipitation from Pluvio, Parsivel, and LPM from 10 precipitating days. The yellow line is the one-to-one relation.

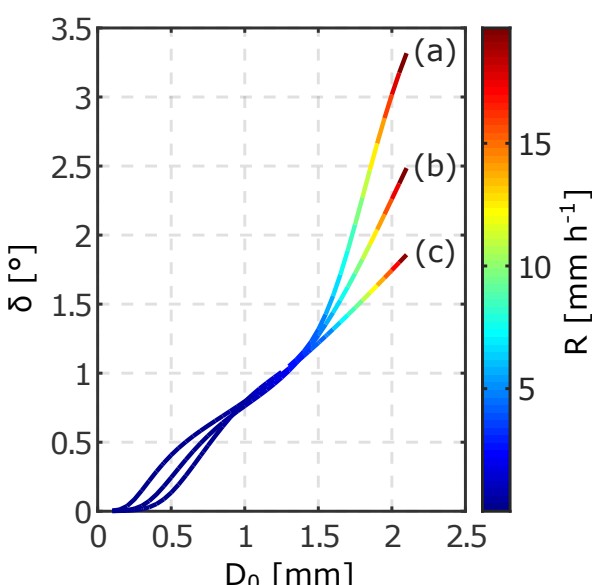

**Figure 4.** Simulated relations between the backscattering differential phase shift $\delta$ and the median drop diameter $D_0$ at 94 GHz. The curves show simulations for DSDs with $N_L = 2500 \, \text{mm}^{-1} \, \text{m}^{-3}$ and $\mu$ equal to 5 (a), 0 (b), and 15 (c). The corresponding rain rate $R$ is color-coded; maximum rain rate for all simulations is limited to values smaller than 20 mm h$^{-1}$. The calculations are made for 20°C and 30° elevation angle.



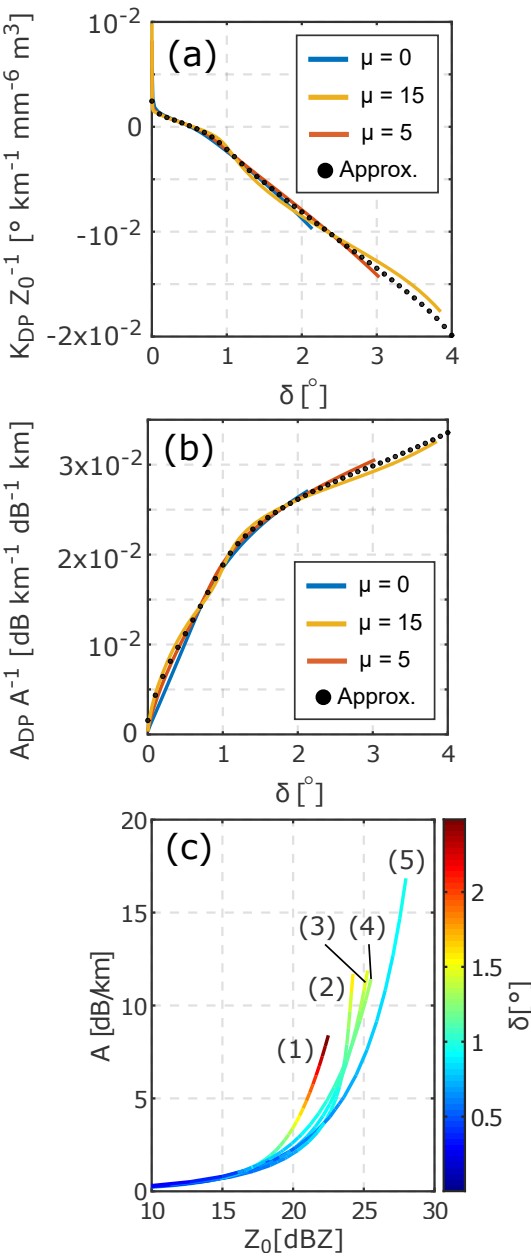

**Figure 5.** Relations between $K_{DP}$, $Z_0$, and $\delta$ (a); $A_{DP}$, $A$, and $\delta$ (b); $A$, $Z_0$, and $\delta$ (c) at 94 GHz. The solid lines show the calculations for different DSD, the black dots denote the fitted approximations (Eq. 1 (a) and Eq. 2 (b)). In panel (c) the calculations are made for $\mu = 5$ and $N_L = 2500$ (d), $\mu = 5$ and $N_L = 8000$ (e), $\mu = 5$ and $N_L = 25000$ (f), $\mu = 0$ and $N_L = 8000$ (g), and $\mu = 15$ and $N_L = 8000$ (h). The rain rates are all smaller than 20 mm h$^{-1}$. All calculations are made for 30° elevation angle and a temperature of 20°C. The units of $Z_0$, $A$, $K_{DP}$, and $A_{DP}$ in the panels (a) and (b) are mm$^6$ m$^{-3}$, dB km$^{-1}$, ° km$^{-1}$, and dB km$^{-1}$, respectively.





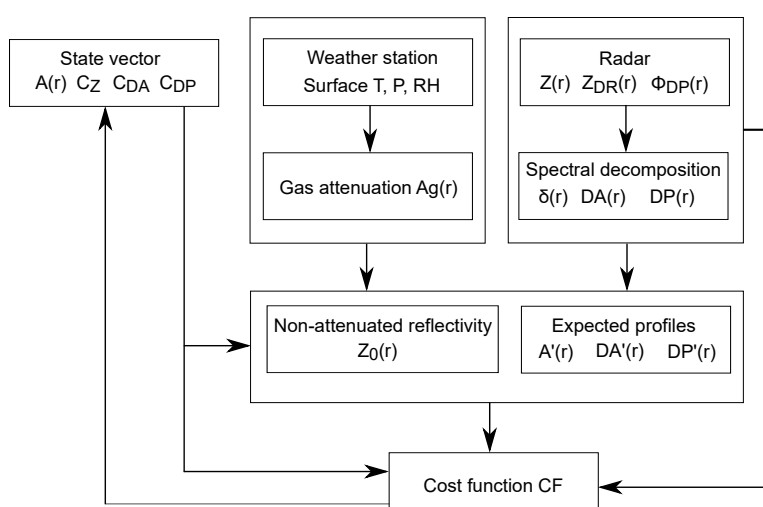

**Figure 6.** Schematic illustrating the processing steps of the self-consistency method. A detailed description can be found in text.



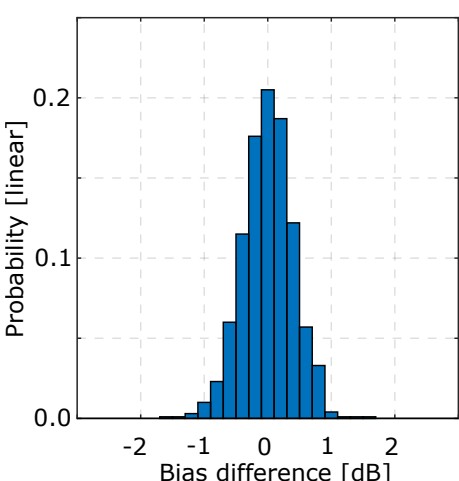

**Figure 7.** Estimated uncertainty of the reflectivity calibration coefficient $C_Z$ using the the self-consistency method (Sec. 3). Various rain DSDs were used to simulate ideal profiles of radar variables. Random noise was then added to these profiles and several calibration coefficients before applying the self-consistency method (see also details in the text). The distribution shows the difference between original $C_Z$ and the retrieved value. The 5th and 95th percentiles, and the standard deviation of the distribution are -0.7, 0.7, and 0.4 dB, respectively.

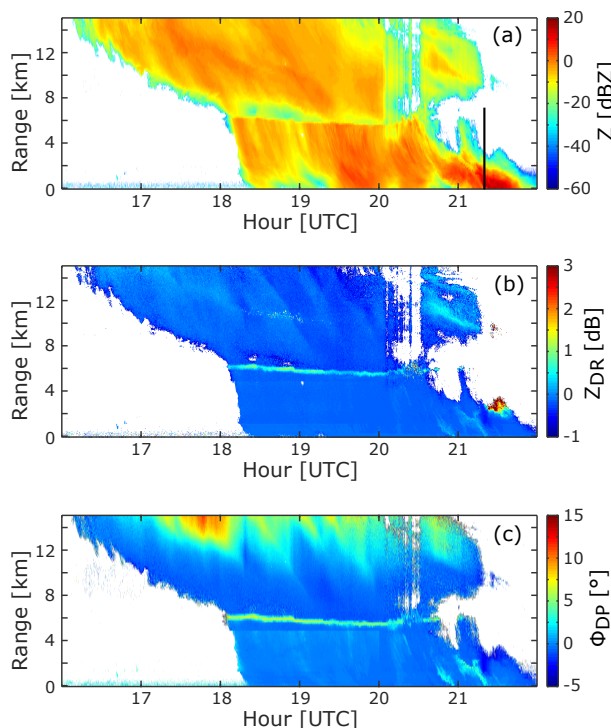

**Figure 8.** Time-range cross sections of radar reflectivity (a), differential reflectivity (b), and differential phase (c). The observations were taken by the radar 2 on 9 June 2018 in Meckenheim, Germany. The radar was pointed to $30°$ elevation. Range corresponds to the slanted distance from the radar. The black vertical line indicates a time sample used for the spectral analysis shown in Fig. 9.



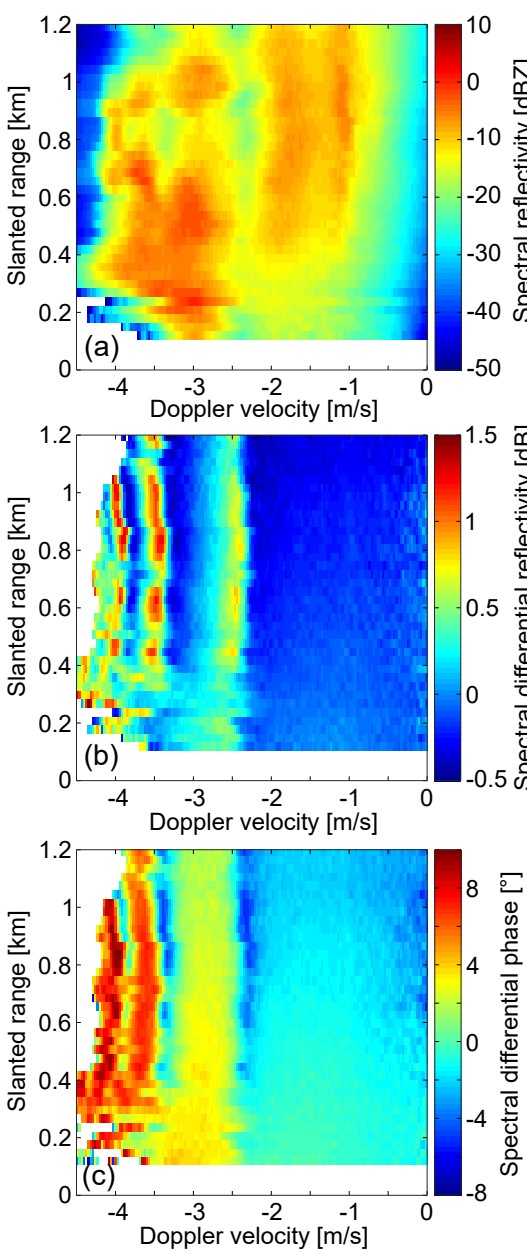

**Figure 9.** Range profiles of Doppler spectra of reflectivity (a), differential reflectivity (b), and differential phase (c). The measurements were taken by the radar 2 on 9 June 2018 at 21:19:23 UTC in Meckenheim, Germany. The radar was pointed to 30° elevation. Negative velocities indicate movements towards the radar. Relatively slow (small) drops are at the right side of the spectrum profile, while the fast falling (big) drops are at the left side. Note that in order to make the figure easier to interpret, the horizontal wind contribution has been roughly mitigated by shifting the right most detected spectral line of a spectrum to 0 m s$^{-1}$.



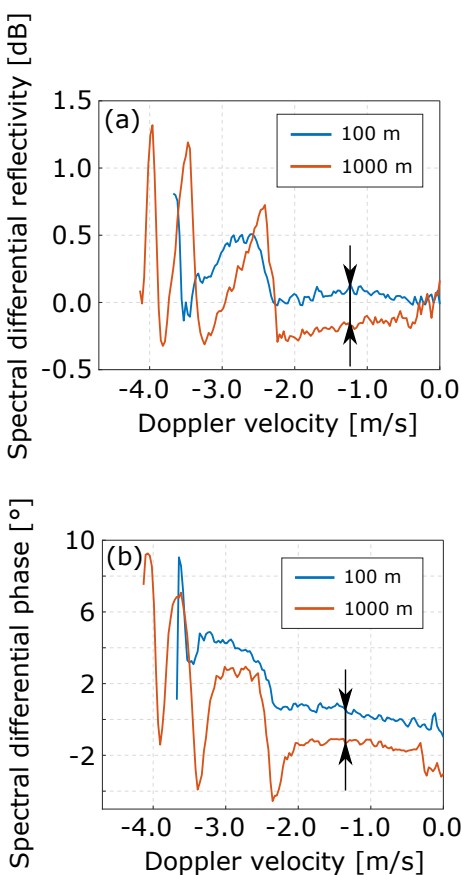

**Figure 10.** Doppler spectra of differential reflectivity (a) and differential phase (b) taken at 0.1 (blue lines) and 1 km (red lines) for the case shown in Fig. 9. The differences between the observations in the area of relatively slowly moving particles indicated by the arrows are associated with the propagation effects, namely differential attenuation in (a) and propagation differential phase shift in (b).

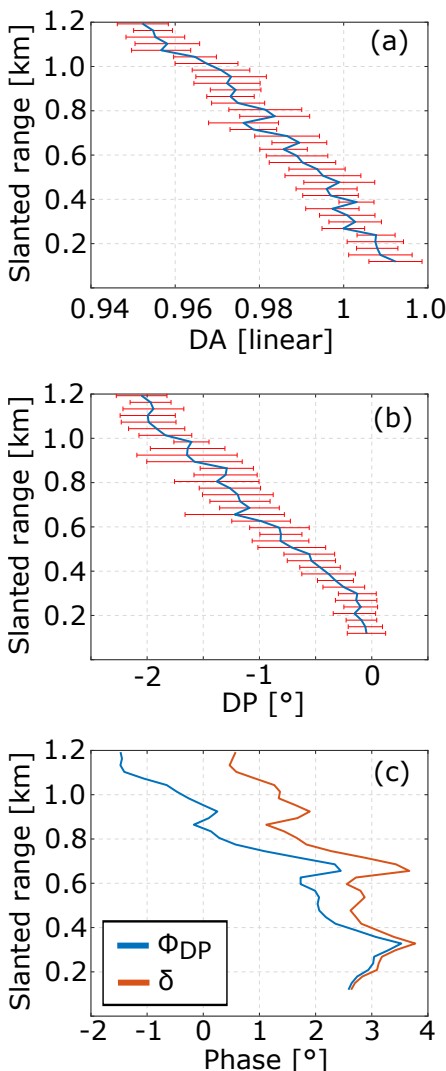

**Figure 11.** Profiles of differential attenuation $DA$ (a) and differential phase $DP$ (b) which are solely due to propagational effects. The profiles have been derived using the spectral decomposition technique illustrated in Fig. 10 applied to the profiles of spectra shown in Fig. 9. Blue lines and red bars indicate mean values and $\pm 1$ standard deviation, respectively. Panel (c) shows profiles of total phase shift $\Phi_{DP}$ (blue) and $\delta$ (red).

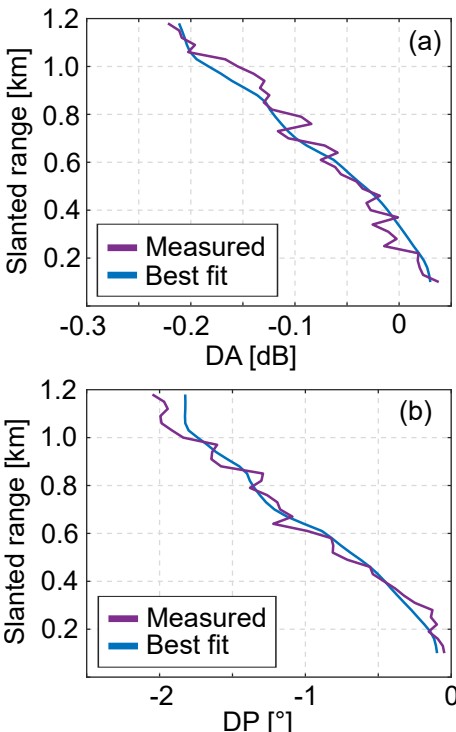

**Figure 12.** Range profiles of DA (a) and DP (b). Blue solid lines correspond to the best fit found by the self-consistency method for radar 2 with $C_Z = -0.7$ dB. The magenta lines represent profiles estimated from the measurements (shown by blue lines in Fig. 11). The results are obtained for the case shown in Figs. 9 and 11.



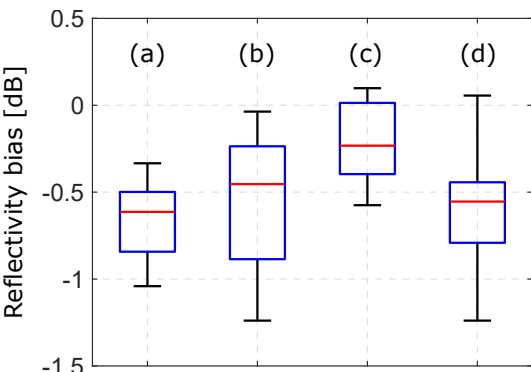

**Figure 13.** Reflectivity biases due to the calibration coefficient $C_Z$ estimated using the self-consistency method applied to observations at $30°$ elevation angle during rain events on 9 June 2018 at 21 UTC (a), 20 July 2018 at 17 UTC (b), and 28 July 2018 at 11:22 UTC (c). The box (d) shows the results for all 64 available samples. The measurements were taken with the radar 2 at the RPG site. The total number of samples are 45 for (a), 11 for (b), and 8 for (c). The upper and lower edges of the boxes correspond to 75th and 25th percentiles, respectively. The upper and lower whiskers indicate 95th and 5th percentiles, respectively. The horizontal red bars correspond to median values.





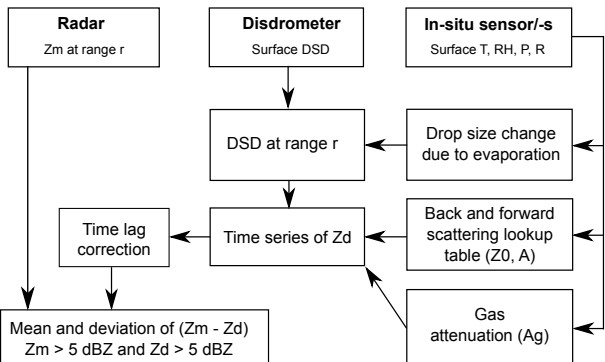

**Figure 14.** Schematic illustration of the extended disdrometer-based method. Detailed descriptions can be found in text.

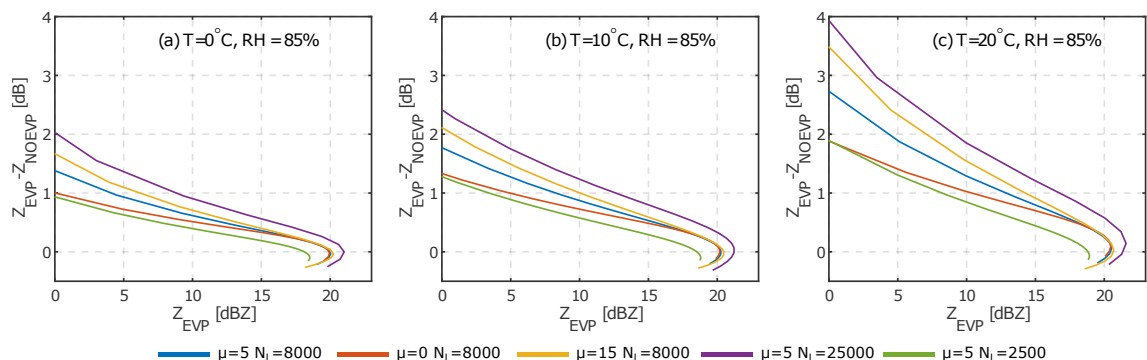

**Figure 15.** Simulated impact of evaporation on the simulated reflectivity at 250 m range for DSDs measured at the surface. $Z_{\mathrm{NOEVP}}$ is the reflectivity at 250 m range calculated with DSDs assumed at the surface without taking evaporation into account. $Z_{\mathrm{EVP}}$ is the reflectivity at 250 m range assumed for the same surface rain DSDs but corrected for drop evaporation within the 250 m layer. For the shown evaporation scenario, surface temperature and humidity are assumed to be $10°\mathrm{C}$ and $85\%$, respectively. Color denotes simulations with different rain DSD parameters.



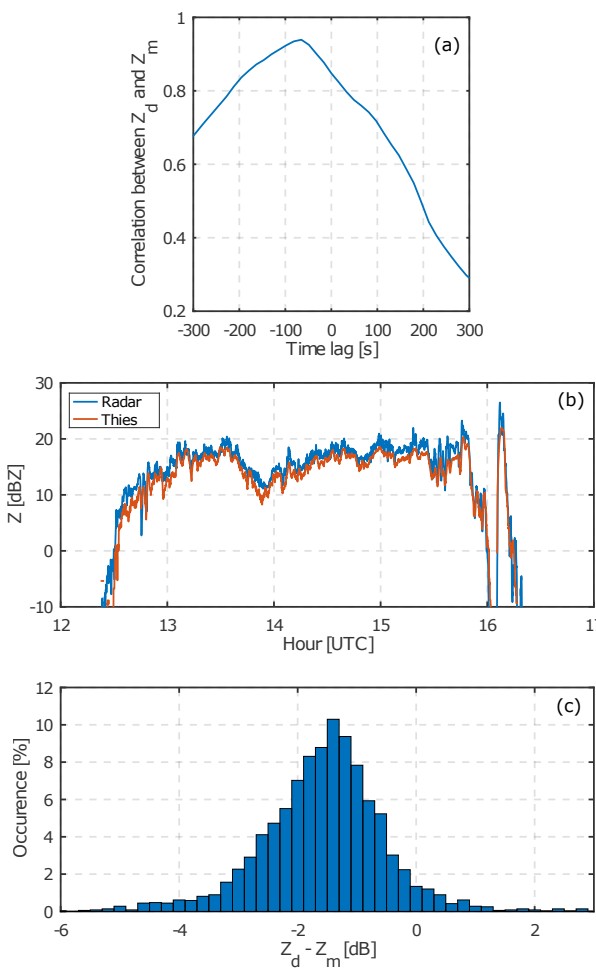

**Figure 16.** Correlation (a) between $Z_d$ and $Z_m$ at 250 m distance from the radar for the rain event on 1 November 2018 observed by radar 1. Various time lags are successively applied in order to detect the most likely time delay (maximum correlation) between 250 m range and surface level. Time series (b) of the reflectivity $Z_m$ measured by the radar 1 at 250 m altitude (blue line) and the reflectivity $Z_d$ modeled from the LPM observations (red line). Panel (c) shows the distribution of $Z_d - Z_m$. Only 3 s samples for which $Z_d$ and $Z_m$ exceed 5 dBZ are used. The mean reflectivity difference is -1.2±0.3 dB. The standard deviation of the mean is calculated according to Appendix C.



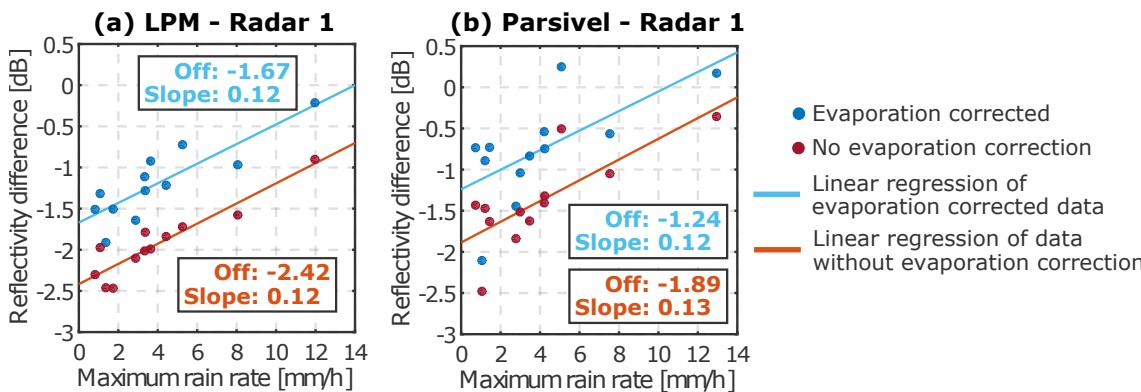

**Figure 17.** The results of the disdrometer-based method from 12 rain events. The calibration evaluation is made for the radar 1 using LPM (a) and Parsivel (b). Each dot represents a result for a single rain event. Solid lines show linear regressions. Offsets and slopes of the regressions are given in corresponding boxes. Color of dots and lines indicates whether the evaporation correction has been applied (blue) or not (red).



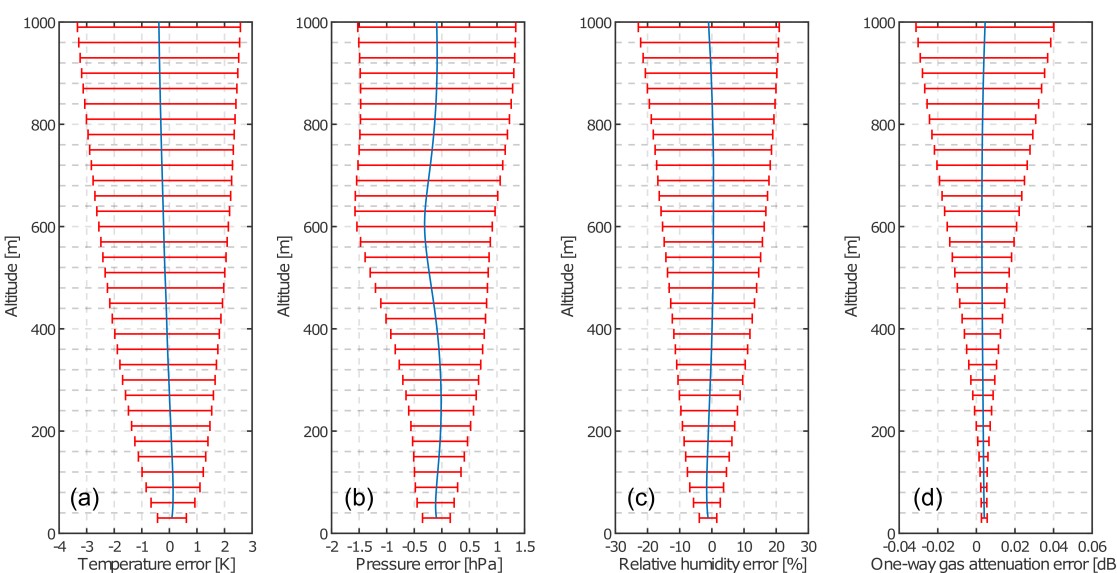

**Figure A1.** Uncertainties in the derived profiles of temperature (a), pressure (b), relative humidity (c), and one-way gas attenuation (d) when using only surface values from a weather station. The uncertainties have been estimated with a large set of radiosonde profiles restricted to a minimum surface relative humidity of 65%. The blue lines and the red bars show mean and $\pm$ one standard deviation of the corresponding difference, respectively.



**Table 1.** Some key technical specifications of the analyzed W-Band radars.

| Parameter | Value |
| --- | --- |
| Center frequency [GHz] | 94 |
| Transmitted power (at antenna output) [W] | 1.5 |
| Antenna type | 2 Cassegrain |
| Antenna gain [dB] | 50.1 |
| Antenna beam width [°] | 0.56 |
| Intermediate frequency range [kHz] | 300 - 3700 |
| Receiver type | homodine |
| System noise figure [dB] | 4.5 |





**Table 2.** Information of type, data periods, and calibration of the radars used for this study.

| No. | Polarimetry | Operation | Radome exchange | Receiver calibration | Transmitter calibration | Observation periods |
|-----|-------------|-----------|-----------------|----------------------|-------------------------|---------------------|
| 1 | Linear depolarization ratio (LDR) mode | Zenith | 1.10.2018 | 25.05.2018, 6.11.2018 | Spring 2018 | 1.11.2018 to 28.02.2019, Jülich |
| 2 | Simultaneous transmission and simultaneous reception (STSR) mode | Scanning | Oct 2018 | 30.05.2018, 6.11.2018 | Spring 2017 | Summer 2018, Meckenheim, 1.11.2018 to 31.01.2019, Jülich |

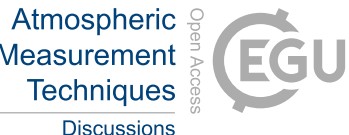

**Table 3.** Ranges of values for differential evolution (DE) used in the self-consistency method.

| Variable | Minimum | Maximum | Units |
|----------|---------|---------|-------|
| $A$ | 0 | 25 | dB km$^{-1}$ |
| $C_Z$ | -6 | 6 | dB |
| $C_{DA}$ | -0.05 | 0.05 | dB |
| $C_{DP}$ | $-0.2$ | 0.2 | ° |



**Table A1.** Coefficients $a_{1-10}$ for the ratio $K_{DP}/Z_0$ in Eq. 1. The coefficients were calculated for $30°$ elevation.

| Index | 0°C | 10°C | 20°C | 30°C |
|---|---|---|---|---|
| 1 | 1.2009 | −1.1258 | −2.8779 | −1.8256 |
| 2 | 0.57645 | 0.40592 | −3.3353 | $3.9987 \times 10^{-3}$ |
| 3 | −0.63757 | −0.29526 | −3.9686 | −0.46699 |
| 4 | 1.1668 | 1.7753 | $6.1885 \times 10^{-4}$ | −3.0455 |
| 5 | −0.58585 | 0.47124 | 5.6427 | −1.2881 |
| 6 | 0.65202 | −0.39622 | −2.9301 | 1.2233 |
| 7 | 1.1906 | 0.70655 | 3.1233 | −3.0428 |
| 8 | $−1.2823 \times 10^{-2}$ | −0.54765 | $−1.4422 \times 10^{-2}$ | 1.2892 |
| 9 | 0.21614 | 0.51483 | −1.6498 | −1.2249 |
| 10 | −0.24463 | $1.2125 \times 10^{-2}$ | $2.6475 \times 10^{-2}$ | −0.79335 |





**Table A2.** Coefficients $b_{1-10}$ for the ratio $A_{DP}/A$ in Eq. 2. The coefficients were calculated for $30°$ elevation.

| Index | 0°C | 10°C | 20°C | 30°C |
|---|---|---|---|---|
| 1 | $-8.2211 \times 10^{-3}$ | $-4.5595 \times 10^{-2}$ | $1.794 \times 10^{-3}$ | 1.9692 |
| 2 | 1.2129 | $-0.77215$ | 3.348 | $-2.1116$ |
| 3 | 0.53705 | 0.86755 | $-2.9703$ | 0.87381 |
| 4 | 10.658 | $-3.3521 \times 10^{-2}$ | 0.48458 | 1.9814 |
| 5 | $7.5556 \times 10^{-2}$ | 0.87027 | 0.49678 | 2.105 |
| 6 | 1.2967 | $-1.2025$ | 1.9062 | $-0.87086$ |
| 7 | 1.4359 | $-0.21182$ | $4.9308 \times 10^{-2}$ | $-3.6404$ |
| 8 | $-0.13421$ | $-1.2072 \times 10^{-2}$ | 0.23934 | $-1.1534 \times 10^{-3}$ |
| 9 | $-0.19654$ | $3.9624 \times 10^{-2}$ | $-2.1627$ | $-3.2384 \times 10^{-2}$ |
| 10 | $-8.8884$ | $1.6606 \times 10^{-2}$ | $-0.41128$ | $-0.1098$ |



**Table A3.** Coefficients $c_{1-17}$ for $A$ in Eq. 3. The coefficients were calculated for $30°$ elevation.

| Index | 0°C | 10°C | 20°C | 30°C |
|---|---|---|---|---|
| 1 | 40.205 | 219.03 | –59.016 | 284.28 |
| 2 | $-7.0826\times10^{-2}$ | 0.23509 | $5.1985\times10^{-2}$ | 0.20864 |
| 3 | –2.3107 | $4.7215\times10^{-5}$ | 2.3352 | $8.2423\times10^{-5}$ |
| 4 | –0.57936 | –0.33685 | 0.58912 | –0.3047 |
| 5 | 97.864 | 135.85 | –194.27 | 130.09 |
| 6 | –0.92476 | –0.24005 | 0.22257 | –0.18727 |
| 7 | $-8.579\times10^{-4}$ | $8.2907\times10^{-4}$ | $-6.3754\times10^{-4}$ | $-7.8938\times10^{-4}$ |
| 8 | 0.57887 | 0.42938 | –0.40174 | $8.4885\times10^{-2}$ |
| 9 | –89.12 | –84.345 | 331.89 | 160.48 |
| 10 | $-8.2716\times10^{-3}$ | 0.2336 | 0.21176 | –0.22374 |
| 11 | $-7.8008\times10^{-4}$ | $1.0573\times10^{-3}$ | $5.3636\times10^{-5}$ | $6.816\times10^{-4}$ |
| 12 | –0.31605 | –0.14118 | –0.30519 | 0.4095 |
| 13 | –96.211 | 9.2789 | –142.34 | 96.122 |
| 14 | –0.93417 | $-8.079\times10^{-2}$ | 0.19708 | $-5.9172\times10^{-2}$ |
| 15 | $-4.8612\times10^{-4}$ | –2.2057 | $7.7241\times10^{-4}$ | –2.2104 |
| 16 | 0.58409 | –0.51589 | –0.1073 | –0.51291 |
| 17 | 12.477 | 13.823 | 68.223 | 107.13 |



**Table A4.** Coefficients for $D_{250}$ in Eq. A6.

| Index | $g$ | $p$ | $q$ | $u$ | $\alpha$ |
|---|---|---|---|---|---|
| 1 | –20.127 | 0.68097 | $-2.4517\times10^{-3}$ | $-7.2329\times10^{-3}$ | 0.86151 |
| 2 | 20.19 | –0.51637 | $1.6484\times10^{-3}$ | $-1.0423\times10^{-3}$ | –0.84634 |
| 3 | –19.996 | $2.8944\times10^{-2}$ | $-1.3688\times10^{-3}$ | $-9.1852\times10^{-3}$ | –0.7242 |
| 4 | –20.054 | –0.1005 | $1.5558\times10^{-3}$ | $8.7377\times10^{-3}$ | 0.26384 |
| 5 | –20.176 | $6.129\times10^{3}$ | $1.3961\times10^{3}$ | $-3.2785\times10^{4}$ | $-3.3515\times10^{2}$ |
| 6 | –19.919 | –0.72646 | $2.2635\times10^{-3}$ | $4.6927\times10^{-3}$ | –0.62472 |
| 7 | 24.614 | $4.5325\times10^{4}$ | $5.0256\times10^{3}$ | $-6.6315\times10^{3}$ | $1.0859\times10^{4}$ |
| 8 | 20.019 | $-4.0408\times10^{-2}$ | $1.0534\times10^{-3}$ | $5.0666\times10^{-3}$ | $9.138\times10^{-2}$ |
| 9 | 19.928 | 0.32405 | $-1.8539\times10^{-3}$ | $-6.4004\times10^{-3}$ | 1.2194 |
| 10 | –24.296 | $-1.6549\times10^{4}$ | $-1.6845\times10^{3}$ | $1.6804\times10^{3}$ | $1.8939\times10^{4}$ |