# Peer review of "Evaluation of the reflectivity calibration of W-band radars based on observations in rain"

_Atmospheric Measurement Techniques, 2020_

## Referee Comment (RC1) · Anonymous Referee #1 · 13 Jun 2020

AMT Review of "Evaluation of the reflectivity calibration of W-band radars based on observations in rain" by A. Myagkov, S. Kneifel and T. Rose.

**General comments**

This article proposes a self-consistency methodology to assess the calibration of the W-band radar. That is the main focus of the paper. For mm-wavelengths, this methodology increases in complexity compared to cm-wavelengths because of Mie scattering effects and attenuation. Next another calibration evaluation technique, which uses disdrometer data, is discussed. This technique is improved by taking into account possible evaporation in the path range bin – disdrometer location. Comparing both methods, consistency is found in radar constant offsets of two W-band radars.

For readers interested in the topic of calibration of W-band radars, this article is very valuable. Further it is very well referenced. Therefore, I recommend this article for publication. In the section "Specific comments", I have some questions to the authors and some corrections for improving the paper.

**Specific comments**

**1)** The spectral polarimetric measurements are acquired at the elevation 30 deg. Why this choice? Is it an optimum elevation angle for the proposed self-consistency method?

**2)** In page 7, the ratios (1)-(2) and the specific attenuation (3) are parameterized as a function of the backscattering differential phase, $\delta$, and, $\delta$ and the measured equivalent reflectivity factor $Z_0$, respectively. Can you discuss the choice of the function $f$ for the parameterization? And for the number of coefficients ($a_i$, $b_i$, $c_i$) where $i$ varies from 1 to 10, or 1 to 17. What is/ are the criterium/ criteria to select these numbers?

**3)** Equations corrections

Replace $S_{jk}(D_i)$ by $F_{jk}(D_i)$ in equations (A4), (B3) and (B4)

$$A_i = 8.686 \times 10^3 \left( \frac{2\pi}{k} \operatorname{Im}\left[ F_{hh}(D_i) \right] \right) \qquad \text{(A4)}$$

$$A_{DP} = 8.686 \times 10^3 \frac{2\pi}{k} \sum_{i=1}^{n} \operatorname{Im}\left[ F_{hh}(D_i) - F_{vv}(D_i) \right] N_i \qquad \text{(B3)}$$

$$K_{DP} = 10^3 \frac{180}{\pi} \frac{2\pi}{k} \sum_{i=1}^{n} \operatorname{Re}\left[ F_{hh}(D_i) - F_{vv}(D_i) \right] N_i \qquad \text{(B4)}$$

In equation (B4), add $10^3$ to express $K_{DP}$ in $^\circ$ km$^{-1}$.

In (B2), $S_{hh}$ is complex conjugate instead of $S_{vv}$.

$$\delta = \frac{180}{\pi} \arg\left[ \sum_{i=1}^{n} N_i S_{hh}^*(D_i) S_{vv}(D_i) \right] \qquad \text{(B2)}$$

**4)** Do the authors use the estimated radar calibration constant, $C_Z$, to correct the equivalent reflectivity factor values of the radar 2?

**5)** Page13, lines 10 and 21: For the calculation of $Z_0$ (equivalently $Z_d$) using the disdrometer DSDs, I don't understand the use of Table A1, which is related to the other method, self-consistency. I may miss the point here.

**6)** For comparing disdrometer and radar reflectivity factor measurements, the trade-off radar range is 250 m. What is the expected underestimation of $Z_m$ related to the FMCW measurement mode at this range? Is this significant for the radar calibration constant?

**7)** What is the argument for the rain rate upper limit 20 mm h$^{-1}$?

**8)** Be consistent with the terminology for the radar observables in the whole paper.

I would avoid the term "shift" for $\Phi_{DP}$, $K_{DP}$ and $\delta$.

Instead of $\Phi_{DP}$ differential phase shift, just term it as "differential phase". Example: line 10 in page 19.

Instead of $K_{DP}$ specific differential phase shift, just term it as "specific differential phase". Example: line 9 in page 15.

Instead of $\delta$ backscattering phase shift, term it as "backscattering differential phase". Examples: line 3 in page 7, line 8 in page 15.

**9)** Page 10, lines 26-28: Taking into account that signal-to-noise ratio in rain within the first kilometer typically exceeds 30 dB, and the copolar correlation coefficient in rain approaches 1, variability in the polarimetric variables are low…….

**10)** Page 11, line 15: discussion related to Fig. 8, … Around 21 UTC positive and negative values in both $Z_{DR}$ and $\Phi_{DP}$ are visible…….I don't see this. Add in Fig. 8 a zoomed window in the area of interest.

**11)** Page 12, line10: …………. (3) the median $K_{DP}$ must be lower than -0.3°km$^{-1}$, and (4) the median $A_{DP}$ is lower than -0.06 dB km$^{-1}$. How are found these threshold values?

**12)** About Table A1.

Mention the units of $N_i$, $V_i$, $v_i$ and $S_i$.

$|K|^2$ is the dielectric factor of water at a certain temperature. How is defined $|K_0|^2 = 0.74$ (water? which temperature?)

Typo in the Table: Parsivel

**13)** What is the meaning of the bending of the curve $Z_{EVP}$-$Z_{NOEVP}$ versus $Z_{EVP}$ at values of $Z_{EVP}$ near 20 dBZ in Figure 15?

**Technical corrections**

**1)** Page 4, line 10: …The calibration methods and their comparison are shown in **Secs. 3-4**.

**2)** Page 8, line 3: …….and $c_{1\text{-}17}$ are given ………

**3)** Replace $\Phi$ by $\Phi_{DP}$ in the whole text (line 12 in page 8, lines 11, 13, 23, 31 in page 11)

**4)** Page 10, line 23: …Size distributions with $A$ less than 3 dB km$^{-1}$ were excluded from the analysis ……. Is it not 0.3 dB km$^{-1}$ instead of 3 dB km$^{-1}$?

**5)** Page 11, line 11: …The melting layer can be depicted at **the height** 2.5 km by enhanced values of ….

**6)** Page 13, line 27, ….to LPM, **Parsivel**, and radar 1 …….

**7)** Page 14, line 1: …The blue dots were calculated according to Sec. 4**.1**, while ….

**8)** Page 14, line 21: …As it was mentioned in Sec. **4.5**, the ….

**9)** Page 15, line 6: …. spectral polarimetry obtained **from** a W-Band radar….

**10)** Page 15, line 11: …based on realistical **assumptions** of errors ………………….

**11)** Page 18, line 13: ……the application of such evaporation correction…….

**12)** Page 18, lines 23-24: …one-way differential attenuation $A_{DP}$ [dB km$^{-1}$], and specific differential phase $K_{DP}$ [° km$^{-1}$]…

**13)** Figure 5 caption: replace (d), (e), (f), (g) and (h) by (1), (2), (3), (4) and (5) to be consistent with Figure 5c.

---

## Referee Comment (RC2) · Anonymous Referee #2 · 21 Jul 2020

Summary ——— This paper looks into the calibration of W-band FMCW cloud radars using a variety of methods. The primary contribution uses self consistency and polarimetric spectral data to select a region of the spectra that is believed to be primarily rayleigh scattering, and use this for the calibration by fitting several self consistency curves. There are also comparisons to using disdrometer scattering as a calibration source. Overall the paper is well written, covers an interesting subject, ande has an actual need in the field. I'm recommending major revision as I believe the handling of the uncertainty could be better, and I have some concerns about the effect of other confounding variables on the accuracy. The paper is however important, and the methodology does appear sound. Overall it was a pleasure to read the paper and I believe after addressing my issues this paper will be an important contribution to the field.

[Figure]

I apologize to the authors for the delay in the review as Covid has mixed a few things up.

Major Comments ———- Section 3.2: One large question I have here is how this varies by temperature, DSR, and choice of canting angle distribution. You discuss in the appendix how it is fairly stable with rotation (and so likely not dependent on DSR and canting angle) but that is only for the back scattering coefficients. Is this true for the forward scattering coefficients and delta upon which this method relies?

The choice of fitting functions (both parametric form, and the actual form of f(x) feels incredibly arbitrary and over parameterized on first read. I'm sure a lot of thought went into this, so maybe a sentence or two justification on why this form and so many parameters over something like a polynomial or power law?

Looking at the fit parameters they seem to vary fairly drastically based on temperature (for instance changing both in order of magnitudes, and in sign).

Section 3.3: There seems to be no discussion of air effects on the fall speed here. Is the argument just that at ∼240 meters there is no vertical air motion, or that it is bounded such that it won't effect the choice of 0-2 m/s choice for spectra? This assumption should be stated. If it is not the case, it should be shown that realistic wind speeds at this height don't effect the methodology.

Section 3.5: I am okay with most of the uncertainty characterization, but page 11 first paragraph uses a value of 0.5 standard deviation for the separation of delta and DP. This feels a little low to me, but can the authors provide some justification for these values?

Minor comments ———- p3.10: "Calibration with a point target does not take into account the volumetric scattering" -> I don't understand this point, nor do I think it matters. The calibration process is only concerned with transmitted/retrieved power and as long as the IF filters are set appropriately, volumetric vs point does not matter.

p6.4: I think you mean 50% of terminal fall speed. The wording behind a factor of 2 is a little ambiguous (for instance 5 m/s , a factor of 2 would mean you reject anything between 0 and 10 m/s)

p7.20: You should define Z0 here by name at least. I know it is done in the appendix, but it took a little bit to track down.

Eq(7): I assume the two dielectric terms are just to account for differences in assumed dielectric at the radar vs the actual measured based on temperature. A sentence should be added just to clarify this.

Section 3.6: Applications to radar 2-> Did I miss applications to radar 1? Later on you bring up radar 1, but maybe change how you refer to them as it is a bit confusing to start with radar 2 in evaluation.

P12.10 The hyphens to stand off the 0.3 deg and 0.06km should be removed, it reads as negative values.

Lapse rate is mispelled as laps rate.

P11.28: Convoluted should be convolved.

---

## Author Comment (AC1) · 4 Aug 2020

The authors would like to thank the two reviewers for the valuable comments, which helped us to improve the manuscript. Below we address all the reviewer's points. The reviewer comments are in blue color, our responses are in black. References to figures and sections are as they were in the original manuscript.

**Reply to Reviewer #1**

1) The spectral polarimetric measurements are acquired at the elevation 30 deg. Why this choice? Is it an optimum elevation angle for the proposed self-consistency method?

Actually, there is no solid argument for exactly 30 deg elevation. We just considered that in the case of zenith pointing (90 deg elevation) the Doppler resolution is the best, because the terminal-velocity projection on the radar-line-of-sight is the largest. Nevertheless, scattering from near-horizontally aligned spheroids is close to isotropic and ZDR and delta are close to 0. In contrast, at 0 deg elevation, polarimetric signatures are the strongest, but the projection of the terminal velocity on the radar beam is close to 0, so Doppler observations are not really possible. We did not optimize the elevation angle for this study and used 30 deg because we have had measurements at this elevation angle. This information was added to the manuscript (Sec. 3.6).

2) In page 7, the ratios (1)-(2) and the specific attenuation (3) are parameterized as a function of the backscattering differential phase, δ, and, δ and the measured equivalent reflectivity factor Z0, respectively. Can you discuss the choice of the function f for the parameterization? And for the number of coefficients (ai, bi, ci) where i varies from 1 to 10, or 1 to 17. What is/ are the criterium/ criteria to select these numbers?

For the function approximation, we typically use non-linear regressions (neural networks). In general, an advantage of neural networks is that they can approximate any function with a number of input/output parameter with a good quality. Since we have this standard tool in hand, we do not have to choose which type of polynomial or power law to choose for different functions. The resulting fits (Eqs. 1-3) are nothing but neural networks with 3 neurons in the hidden layer. Since the explanation of the neural networks would add another section to already quite lengthy manuscript we just provided a final result in form of equations/tables and ready-to-use functions (in supplementary materials, so a reader does not need to type all these coefficients manually in his/her code). The quality of the fits is discussed in the text. We agree that another fitting strategy could lead to similar results, but we do not think that this would affect the results. We added to the manuscript that the approximations were obtained using neural networks.

3) Equations corrections
Thanks for checking the formulas. We corrected all the items.

4) Do the authors use the estimated radar calibration constant, CZ, to correct the equivalent reflectivity factor values of the radar 2?

We added a sentence to the summary section: „The derived calibration factors can be used to monitor the radar stability and to correct the observed reflectivity values"

5) Page13, lines 10 and 21: For the calculation of Z0 (equivalently Zd) using the disdrometer DSDs, I don't understand the use of Table A1, which is related to the other method, selfconsistency. I may miss the point here.

Please note, that the non-attenuated reflectivity and attenuation are used in both methods. Therefore, the appendix A is relevant for both methods. The table A1 specifies how to calculate the non-attenuated reflectivity and attenuation using disdrometer data.

6) For comparing disdrometer and radar reflectivity factor measurements, the trade-off radar range is 250 m. What is the expected underestimation of Zm related to the FMCW measurement mode at this range? Is this significant for the radar calibration constant?

The estimated overlap of two beams is about 90% at 250 m. In order to correct for this we apply the method by Sekelsky and Clothiaux (2002). Please note that this information is given in the first paragraph of the Sec. 4: „For a two-antenna system, such as used for the FMCW systems in this study, the incomplete beam overlap is corrected for using the method in Sekelsky and Clothiaux (2002). At ranges larger than 250 m from the radar, the beam overlap for all radars used is better than 90%."

7) What is the argument for the rain rate upper limit 20 mm h-1?

In the case of strong rain, the liquid water on the radomes can affect the calibration results. This is one of the conclusions from Fig. 17 (please see sec 4.4). According to the figure, these effects might become significant (1-2 dB) already at 10 mm/hr. Since the radome effects are hard to quantify, we do not extend the method to higher rain rates.

8) Be consistent with the terminology for the radar observables in the whole paper. I would avoid the term "shift" for $\Phi_{DP}$, $K_{DP}$ and $\delta$. Instead of $\Phi_{DP}$ differential phase shift, just term it as "differential phase". Example: line 10 in page 19. Instead of $K_{DP}$ specific differential phase shift, just term it as "specific differential phase". Example: line 9 in page 15. Instead of $\delta$ backscattering phase shift, term it as "backscattering differential phase". Examples: line 3 in page 7, line 8 in page 15.

Corrected

9) Page 10, lines 26-28: Taking into account that signal-to-noise ratio in rain within the first kilometer typically exceeds 30 dB, and the copolar correlation coefficient in rain approaches 1, variability in the polarimetric variables are low.......

Corrected

10) Page 11, line 15: discussion related to Fig. 8, ... Around 21 UTC positive and negative values in both ZDR and $\Phi_{DP}$ are visible.......I don't see this. Add in Fig. 8 a zoomed window in the area of interest.

Added

11) Page 12, line10: ............. (3) the median KDP must be lower than -0.3okm-1, and (4) the median ADP is lower than -0.06 dB km-1. How are found these threshold values?

The self-consistency method relies on propagation variables KDP and ADP, which are used to constrain profiles of Z. In the case that magnitudes of KDP and ADP are low and comparable with measurement noise, the method shows larger uncertainties. In general, the large the magnitude of KDP and ADP the better the Z constraint. Using the approach shown in Sec. 3.5 we identified ranges of the method applicability by keeping the uncertainties in the calibration factor within the accuracy specified in Sec. 3.5. This information was added to the manuscript.

12) About Table A1. Mention the units of Ni, Vi, vi and Si.

Added

|K|2 is the dielectric factor of water at a certain temperature. How is defined |K0|2 = 0.74 (water? which temperature?)

Water at 8 deg C. This is added to the caption of the table A1.

Typo in the Table: Parsivel

Corrected

13) What is the meaning of the bending of the curve ZEVP-ZNOEVP versus ZEVP at values of ZEVP near 20 dBZ in Figure 15?

The value on the x-axis is mainly defined by the non-attenuated reflectivity and attenuation by liquid, which both monotonically increase with rain rate. The attenuation increases with rain rate a bit faster than the non-attenuated reflectivity. This leads to the fact that the same reflectivity value can correspond to two different rain rates (lower and larger). This effect can also be seen in Fig. 2 in the paper by Hogan et al. 2002 (JTECH). In the case of lower rain rate, the contribution of small drops (which evaporate faster) is larger. At higher rain rates, there are more large drops which evaporate slower. This explains why the same value of ZEVP corresponds to two different points on the y-axis. This information is added to the manuscript.

**Technical corrections**

1) Page 4, line 10: ...The calibration methods and their comparison are shown in Secs. 3-4.

Corrected

2) Page 8, line 3: .......and $c_{1-17}$ are given .........

Corrected

3) Replace $\Phi$ by $\Phi_{DP}$ in the whole text (line 12 in page 8, lines 11, 13, 23, 31 in page 11)

Corrected

4) Page 10, line 23: ...Size distributions with A less than 3 dB km-1 were excluded from the analysis ....... Is it not 0.3 dB km-1 instead of 3 dB km-1?

The variable A is specific attenuation (not to be confused with Adp which is differential attenuation). The attenuation by rain at 94 GHz is relatively large. 0.3 dB/km attenuation is characteristic to liquid clouds but it is too low for rain. So, 3 dB/km is right here.

5) Page 11, line 11: ...The melting layer can be depicted at the height 2.5 km by enhanced values of ....

Corrected

6) Page 13, line 27, ....to LPM, Parsivel, and radar 1 .......

Corrected

7) Page 14, line 1: ...The blue dots were calculated according to Sec. 4.1, while ....

Corrected

8) Page 14, line 21: ...As it was mentioned in Sec. 4.5, the ....

I checked the version at the AMT site. It is written exactly this way. It is not clear to us what is the reviewer's concern.

9) Page 15, line 6: …. spectral polarimetry obtained from a W-Band radar….

Corrected

10) Page 15, line 11: …based on realistical assumptions of errors ……………………

Corrected

11) Page 18, line 13: ……the application of such evaporation correction…….

Corrected

12) Page 18, lines 23-24: …one-way differential attenuation ADP [dB km-1], and specific differential phase KDP [ o km-1]…

Corrected

13) Figure 5 caption: replace (d), (e), (f), (g) and (h) by (1), (2), (3), (4) and (5) to be consistent with Figure 5c.

Corrected

**Reply to Reviewer #2**

Major Comments:

1.    Section 3.2: One large question I have here is how this varies by temperature, DSR, and choice of canting angle distribution. You discuss in the appendix how it is fairly stable with rotation (and so likely not dependent on DSR and canting angle) but that is only for the back scattering coefficients. Is this true for the forward scattering coefficients and delta upon which this method relies?

Please note, that in order to take the temperature dependence of the scattering properties of raindrops into account, we provide fit coefficients (used in Eq. 1-3) for temperatures 0, 10, 20, and 30 deg C. The coefficients are given in Appendix A (Tables A1 – A4) of the original manuscript and ready-to-use Octave/MATLAB functions are provided in supplementary materials. We now explicitly mention this in the sec. 3.2.

We are not sure what the reviewer refers to with DSR. We assume that the reviewer means drop-size-distribution. Please also note that the self-consistency method (also the original version from Goddard for cm-wavelength) is based on the relations between backscattering and propagation variables, which are nearly immune to variabilities in drop-size-distribution. As shown in Fig. 5, the relations are calculated for different combinations of the shape parameter, median diameter, and concentration. We used a huge number of combinations of these parameters to simulate radar observables. This is described in the beginning of sec. 3 of the original manuscript:

"To infer suitable relations between radar observables, we simulate them using the T-Matrix model (Mishchenko, 2000) and a range of PSDs similar to Hogan et al. (2003). We assume normalized gamma distributions with μ from 0 to 15 and NL from 5×10^2 to 2.5×10^4 mm−1 m−3 . For given μ and NL, the median volume diameter D0 was increased in 0.05 mm steps starting at 0.1 mm until the rain rate reached 20 mm h−1"

These simulations (several thousands of size-distributions) are used to get the fits (eqs. 1-3). The uncertainties of these fits are given in the end of the section of 3.2. We reformulated a sentence to make it clearer: "Using the large set of **simulated** rain PSDs **introduced above** and **corresponding** forward simulated radar parameters, we can parameterize…"

We agree with the reviewer that analysis of the dependence of delta and propagation parameters on canting angle distribution is missing. We now added a figure showing simulations based on T-matrix scattering model. Simulations were made for 0 deg elevation as one expects the largest effect from drop's canting at the lowest elevation angle. The canting angle was assumed to be in the polarization plane. A number of studies show that observed standard deviation in the canting angle typically does not exceed 10 deg. We thus compared radar variables calculated assuming horizontal alignment with those calculated with 10 deg standard deviation. The results show that the canting angle affects backscattering and specific differential phase. The differences at 20 mm/h may reach 0.5 deg and 0.2 deg/km, respectively. Other parameters are not affected much. The found uncertainties due to the canting angle are within the uncertainty levels assumed in sec. 5.3, and thus do not change the results of our study. This information is added to the sec. 3.5.

2. The choice of fitting functions (both parametric form, and the actual form of f(x) feels incredibly arbitrary and over parameterized on first read. I'm sure a lot of thought went into this, so maybe a sentence or two justification on why this form and so many parameters over something like a polynomial or power law?

For the function approximation, we typically use non-linear regressions (neural networks). In general, an advantage of neural networks is that they can approximate any function with a number of input/output parameter with a good quality. Since we have this standard tool in hand, we do not have to choose which type of polynomial or power law to choose for different functions. The resulting fits (Eqs. 1-3) are, thus, nothing but neural networks with 3 (Eqs. 1 and 2) and 4 (Eq. 3) neurons in the hidden layer. Since the explanation of the neural networks would add another section to already quite lengthy manuscript we just provided a final result in form of equations/tables and ready-to-use functions (in supplementary materials, so a reader does not need to type all these coefficients manually in his/her code). The quality of the fits is discussed in the text. We agree that another fitting strategy could lead to similar approximation quality, but we do not think that this would affect the results.

3. Looking at the fit parameters they seem to vary fairly drastically based on temperature (for instance changing both in order of magnitudes, and in sign).

As mentioned in the previous answer, we use neural network for fitting. Training of neural networks is a stochastic process with random initial conditions. Therefore, two neural networks trained to solve the same task can converge to different sets of coefficient values. There is not much physical meaning in sign and absolute values of the coefficients. What is important is the fit quality, which is appropriate for the method.

4. Section 3.3: There seems to be no discussion of air effects on the fall speed here. Is the argument just that at ~240 meters there is no vertical air motion, or that it is bounded such that it won't effect the choice of 0-2 m/s choice for spectra? This assumption should be stated. If it is not the case, it should be shown that realistic wind speeds at this height don't effect the methodology.

Please note, that for the self-consistency method observations are made at 30 deg elevation. At this relatively low elevation angle in addition to vertical air motions, we have strong contributions from the horizontal wind, which are often much stronger than vertical motions. Due to air motions (regardless vertical or horizontal) the Doppler velocity corresponding to each spectral line is a sum of two components, namely projections of terminal velocity and air motions. In general, air motions would shift the whole spectra along the velocity axis,

but the shape of spectra is still roughly the same in the case of low turbulence. For the described method, the exact knowledge of the terminal velocities is not required. We roughly mitigate air motions (both vertical and horizontal) by shifting spectra is such a way that the right most detected spectral line (drops with the slowest fall velocity) corresponds to 0 m/s. As shown in Sec 3.6, such rough mitigation is good enough to clearly separate drops with Rayleigh scattering and those producing resonance effects. This information was added to the Sec. 3.3.

5. Section 3.5: I am okay with most of the uncertainty characterization, but page 11 first paragraph uses a value of 0.5 standard deviation for the separation of delta and DP. This feels a little low to me, but can the authors provide some justification for these values?

The separation of delta and KDP is done using polarimetric spectra, where Rayleigh and resonance zones are clearly separable. In section 3.6 we show an example, where one can see that the standard deviation of the measured differential phase in the Rayleigh area is typically about 0.2-0.3 deg (Figs. 9c, 10b, and 11b). If we average over all spectral lines within this area, the standard deviation becomes even lower. The value we assume as an uncertainty of the separation (0.5 deg) is therefore already a very conservative assumption and even exceeds the actual measurement uncertainty.

6. Minor comments ——- p3.10: "Calibration with a point target does not take into account the volumetric scattering" -> I don't understand this point, nor do I think it matters. The calibration process is only concerned with transmitted/retrieved power and as long as the IF filters are set appropriately, volumetric vs point does not matter.

We do not agree with the reviewer here. The radar equation for meteorological (volume distributed) targets includes a characterization of the antenna beam pattern often by means of the antenna gain and beam width. A point target covers a very small part of the radar beam and thus cannot characterize the complete antenna pattern. Moreover, if the exact position of the point target in the beam is not known, this may introduce a large uncertainty in such an end-to-end calibration. If a location of the target is just a few 0.1 deg off relative to the maximum gain direction (antenna half power beam-width is 0.56 deg), the antenna gain can easily be lower than the maximum by a few dB. Therefore, an accurate characterization of the whole system including the antenna patterns is crucial. This is exactly what is pointed out by Gorgucchi et al (1992) and Chandrasekar et al. (2015).

7. p6.4: I think you mean 50% of terminal fall speed. The wording behind a factor of 2 is a little ambiguous (for instance 5 m/s , a factor of 2 would mean you reject anything between 0 and 10 m/s)

We reformulated the sentence as follows: 'In order to further reduce the effects of splashing on the calculated rain rate and reflectivity, we follow the approach of Tokay et al. (2014) and reject all particles with velocities outside the range of ±50% relative to a theoretical size-velocity relation (Foote and Du Toit, 1969).

8. p7.20: You should define Z0 here by name at least. I know it is done in the appendix, but it took a little bit to track down.

We added names of all variables in the text: '(non-attenuated reflectivity $Z_0$, one-way attenuation $A$, differential reflectivity $Z_{DR}$, specific differential phase $K_{DP}$, differential attenuation $A_{DP}$, and backscattering phase $\delta$)'

9. Eq(7): I assume the two dielectric terms are just to account for differences in assumed dielectric at the radar vs the actual measured based on temperature. A sentence should be added just to clarify this.

This is correct. We added a sentence: 'The two dielectric factors in Eq.~\ref{eq:z0} account for differences in the actual dielectric properties of liquid water and those assumed in the radar software.'

10. Section 3.6: Applications to radar 2-> Did I miss applications to radar 1? Later on you bring up radar 1, but maybe change how you refer to them as it is a bit confusing to start with radar 2 in evaluation.

We changed 'radar 1' to 'radar 2' and vice versa.

11. P12.10 The hyphens to stand off the 0.3 deg and 0.06km should be removed, it reads as negative values.

Please note, that the value of KDP of -0.3 deg/km is indeed negative here. But the value of the differential attenuation should be positive. We corrected this. Similar problem was in sec. 5. Where KDP of +0.3 deg/km was given while it should be negative. We rephrased the sentence in sec. 5: "The differential attenuation and specific differential phase shift should preferably be higher than 0.06 dB km−1 and lower than −0.3 km−1, respectively."

12. Lapse rate is mispelled as laps rate.

Corrected

13. P11.28: Convoluted should be convolved

Corrected

---

## Author Response (AR2)

Dear editor,

Thanks a lot for your valuable comments. Below we address each of item. The comments are given in blue color. Our response is in black.

1. You use a neural network to describe the relationship between propagation and backscattering variables (section 3.1). Only the RMSE is mentioned in the text, and I strongly recommend to add more information (you have thousands of points and simple networks, so limited risks of overfitting, but such aspects - learning curves for instance - should still be provided to the reader...).

Our aim was to give a reader a final approximation results. In the very beginning we thinking about explaining all details of the neural network approach. Such explanation would be quite lengthy and we finally decided not to include it, taking into account that the manuscript is already long. In addition neural network are not the central part of the calibration algorithm, as the reviewers noted, the approximation could probably be done in a simpler way. We used neural networks because we have this tool in hands and it is not necessary to decide which function to choose for the fit (exponential, polynomial etc.).

Please note, that Figure 5 illustrates the neural network approximation results. In panels a and b we show in colored lines some of the data used as input, while the dotted line shows the neural network approximation. The panel c shows only the approximation. All panels show that the approximations are smooth curves which is an indicator of a good generalization, i.e. that the results are not affected by the overfitting effects.

In order to avoid the overfitting we used two approaches: (1), as the editor mentioned, we have chosen a low number of neuron in the hidden layer, (2) we used the Bayesian regularization, which restricts the magnitudes of the weights. In this approach we minimize the cost function $F = a*Ed + b*Ew$, where $Ed$ is a sum of squared errors and $Ew$ is a sum of squared weights. A and b are coefficients inversely proportional to the variances or the weights and errors, respectively. The Bayesian algorithm automatically estimates the coefficients a and b at each iteration.

Learning curves are typically plotted for trainings with the early stopping criterion. In this case the dataset is split into training and validation parts. If the squared error calculated from the training dataset becomes the same as the one calculated from the validation dataset the training is terminated. If not terminated the error on the training dataset can go further down while on the validation dataset it stays the same or even increase. Please note that we do not use this approach for the training.

MacKay1992 and Hagan2014 state that the training with the Bayesian regularization is robust against overfitting and shows similar results as the early stopping criterion. Please note that the learning curves are not required in the case of the Bayesian regularization to inspect the neural network quality. We included a more detailed description of the neural network training (Bayesian approach, Levenberg-Marquardt minimization, and reasoning for these methods). In addition, we added correlations between the target values and the approximations, which are for all variables ~0.998.

2. Regarding item 5 of Reviewer 2, I would recommend to add in the manuscript (end of new Section 3.5) the reference to figures 9,10 and 11 to justify the value of 0.5 deg for the noise in DP (as was done in the responses).

This information was added to section 3.5

[revised manuscript text omitted]